# EPICACHE: EPISODIC KV CACHE MANAGEMENT FOR LONG CONVERSATIONAL QUESTION ANSWERING

## ABSTRACT

Modern large language models (LLMs) extend context lengths to up to millions of tokens, enabling AI assistants to generate coherent and personalized responses grounded in long conversational histories. This ability, however, hinges on Key-Value (KV) caching, whose memory grows linearly with dialogue length and quickly becomes the bottleneck in resource-constrained environments. An active line of research for reducing memory bottleneck is KV cache compression, which seeks to limit cache size while preserving accuracy. Yet existing methods face two major limitations: (i) evicting the KV cache after full-context prefill causes unbounded peak memory, and (ii) query-dependent eviction narrows the cache to a single query, leading to failure cases in multi-turn conversations. We introduce EPICACHE, a training-free KV cache management framework for long conversational question answering (LongConvQA) under fixed memory budgets. EPICACHE bounds cache growth through block-wise prefill and preserves topic-relevant context via episodic KV compression, which clusters conversation history into coherent episodes and applies episode-specific KV cache eviction. We further design an adaptive layer-wise budget allocation strategy that measures each layer's sensitivity to eviction and distributes the memory budget across layers accordingly. Across three LongConvQA benchmarks, EPICACHE improves accuracy by up to 40% over recent baselines, sustains near-full KV accuracy under 4–6× compression, and reduces latency and memory by up to $2.4\times$ and $3.5\times$, thereby enabling efficient multi-turn interaction under strict resource constraints.

## 1 INTRODUCTION

Large language models (LLMs) (Brown et al., 2020; Yang et al., 2025; Touvron et al., 2023; Jiang et al., 2023) have significantly extended their context lengths, with LLM-based AI assistants now capable of processing millions of tokens (Reid et al., 2024; Meta, 2025). This capability enables assistants to leverage extensive dialogue histories when generating responses, producing personalized and contextually coherent outputs (OpenAI, 2024; Anthropic, 2024), which are central requirements for conversational AI applications (Fu et al., 2022).

Long Conversational Question Answering (LongConvQA) is the task of answering a sequence of user questions grounded in extended interaction histories, where sessions span hundreds of turns and multi-session interactions unfold over days or weeks. Recent work has formalized LongConvQA in both human-human conversations and user-AI assistant interactions (Maharana et al., 2024; Lee et al., 2025; Wu et al., 2025a). While most current systems rely on external memory modules (Zhong et al., 2024; Chhikara et al., 2025), the challenge of sustaining such long contexts under strict memory budgets remains largely unaddressed.

To bridge this gap, we study how to enable LongConvQA under constrained memory by designing a Key-Value (KV) cache control framework. The KV cache stores the Key and Value states of each token for reuse in auto-regressive generation, but its size grows linearly with context length, creating severe challenges in extended conversations. For instance, in multi-day dialogues between user and assistant (Wu et al., 2025a), the KV cache of LLaMA3.2-3B exceeds 7GB after only 30 sessions—larger than the size of the model parameters. This underscores the importance of cache management for deploying conversational AI system under resource-constrained environments.

Prior work has attempted to mitigate the growing memory footprint of KV caches in long conversations through various compression techniques (Zhang et al., 2023; Li et al., 2024; Cai et al., 2025). Yet two major limitations remain under memory-constrained settings. First, most methods apply compression after prefilling the entire input context (*post-prefill*), causing peak memory usage that scales linearly with input length. Second, *query-dependent* eviction (Li et al., 2024) retains cache entries by focusing on the current query, and neglects information needed for future queries, thereby degrading accuracy in multi-turn conversations.

We propose EPICACHE, a training-free KV cache management framework that enforces a constant memory footprint through block-wise prefill. After processing each block, we evict less critical KV entries to free space for the next block, ensuring memory consumption remains bounded. Based on block prefill, EPICACHE incorporates an episodic clustering method inspired by conversation segmentation studies (Joty et al., 2013; Galley et al., 2003). Specifically, we apply semantic clustering to group conversation history into coherent episodes, and perform episodic KV cache compression, yielding topic-specific caches while using constrained memory.

Finally, we find that LLMs exhibit different sensitivities to block prefill eviction across layers. Building on this observation, we propose an adaptive layer-wise budget allocation strategy that distributes KV cache budget proportionally to each layer's sensitivity. Together with episodic eviction, this enables EPICACHE to preserve long-range conversational context while operating under strict memory limits, yielding up to 40% higher scores than recent baselines and sustaining accuracy comparable to full KV under $4$–$6\times$ cache compression. In addition, our block-wise cache control framework reduces peak memory usage by up to $3.5\times$, while cache eviction accelerates decoding, cutting latency by as much as $2.4\times$ compared to full KV.

## 2 BACKGROUND

We begin by formalizing Long Conversational Question-Answering (LongConvQA) in Section 2.1. We then discuss memory-constrained KV cache management in Section 2.2, comparing post- and block-prefill eviction and discussing the resulting accuracy-memory trade-offs. Finally, in Section 2.3, we review attention-guided cache compression with patched prompts and present analyses that motivate our method, EPICACHE.

### 2.1 LONG CONVERSATIONAL QA FORMULATION

We formalize LongConvQA as answering a sequence of user queries $\mathcal{Q} = \{q_1, \ldots, q_{N_q}\}$, with $N_q$ denoting the total number of queries, conditioned on long conversational history (Maharana et al., 2024; Lee et al., 2025; Wu et al., 2025a). Let the dialogue history be represented as an ordered sequence of $N_u$ utterances where each utterance $u_j$ pairs a role $r_j$ with text $t_j$:

$$\mathcal{H} = \{u_1, u_2, \ldots, u_{N_u}\}, \quad u_j = (r_j, t_j), \ r_j \in \{\text{speaker}_1, \text{speaker}_2\}, \tag{1}$$

Given a long conversation $\mathcal{H}$, an LLM encodes it into a Key-Value (KV) cache $\text{KV}_{\mathcal{H}}$. For $L$ layers and $H$ KV heads, encoding $N$ tokens produces $L \times H \times N$ KV entries, growing linearly with the conversation length. In this work, we focus on token-level cache compression, where KV entries of less important tokens are evicted; the resulting compressed cache is denoted as $\widetilde{\text{KV}}_{\mathcal{H}} \subseteq \text{KV}_{\mathcal{H}}$, and we use the terms compression and eviction interchangeably.

Our goal is to generate accurate answers for all queries $q_1, \ldots, q_{N_q}$ grounded in the dialogue history $\mathcal{H}$, using a compressed cache $\widetilde{\text{KV}}_{\mathcal{H}}$ that satisfies a memory budget $M$, i.e., with size $L \times H \times M$, while preserving answers comparable to full KV cache ($\text{KV}_{\mathcal{H}}$) based generation:

$$f_{\text{LM}}(q_i \mid \widetilde{\text{KV}}_{\mathcal{H}}) \approx f_{\text{LM}}(q_i \mid \text{KV}_{\mathcal{H}}), \quad i = 1, \ldots, N_q. \tag{2}$$

This formulation serves as an evaluation of *multi-turn conversation* accuracy, where multiple query-answer pairs are grounded in the same dialogue history.

### 2.2 KV CACHE MANAGEMENT: POST PREFILL VS BLOCK PREFILL

Most existing KV compression approaches reduce cache size in the decoding stage by performing eviction *after* the full context has been prefilled, i.e., *Post Prefill Eviction* (Li et al., 2024; Feng et al.,

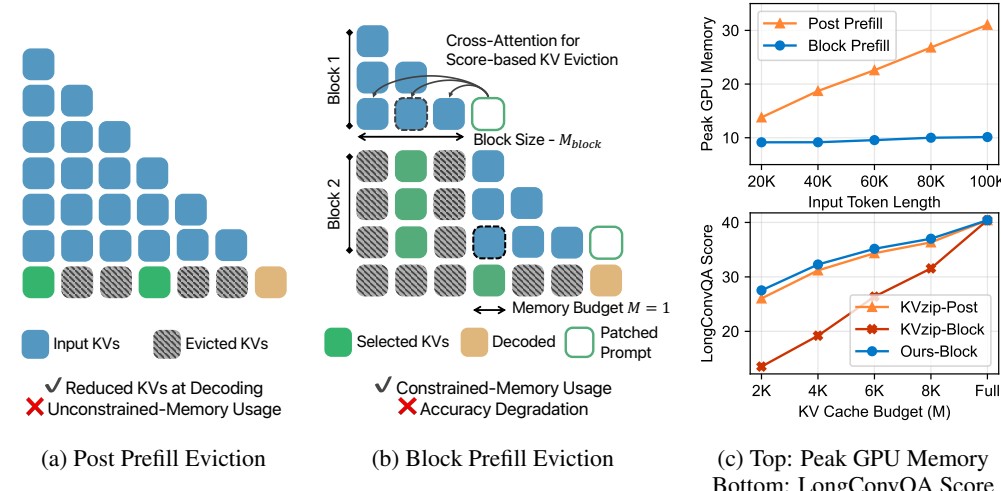

(a) Post Prefill Eviction  (b) Block Prefill Eviction  (c) Top: Peak GPU Memory  
Bottom: LongConvQA Score

Figure 1: **KV Cache Management Analysis.** (a) Post prefill eviction: eviction after full-context prefill, reducing KV size at decoding but causing unbounded memory usage. (b) Block prefill eviction: input processed in 3-token blocks with patched prompts for scoring, then evicted to 1 token. (c) Top: Peak GPU memory vs. input length on LLaMA-3.2-3B with A100. Bottom: LongConvQA accuracy of KV compression methods under post vs. block prefill on LLaMA-3.2-3B.

2024; Cai et al., 2025; Kim et al., 2025). As shown in Figure 1a, this design causes peak memory usage to grow linearly with input length, since the entire context must be cached before any eviction takes place. With optimized attention kernels (Dao, 2024), the prefill stage remains unbounded in memory demand, as observed in Figure 1c top.

To bound memory growth, *Block Prefill Eviction* (Kim et al., 2024; Corallo & Papotti, 2024; Park et al., 2025) processes the input in a block-wise way, handling one segment at a time under a fixed budget $M$. Each step adds $M_{\text{block}}$ tokens, after which eviction reduces KV cache entries back to $M$. For example, in Figure 1b, the budget is $M = 1$, and each block adds $M_{\text{block}} = 3$ tokens that are evicted to $M$. This design ensures the number of cache entries never exceeds $M + M_{\text{block}}$, keeping peak GPU memory usage flat with input length as highlighted in Figure 1c top.

However, this bounded memory comes with a steep accuracy trade-off: when the state-of-the-art post prefill eviction method KVzip, (Kim et al., 2025; NVIDIA, 2025) is applied in the block prefill setting, LongConvQA scores (Maharana et al., 2024) degrade sharply across all budget levels. This underscores a central challenge—while block prefill guarantees constant memory usage, adapting post-prefill eviction methods to this setting severely undermines answer quality in LongConvQA.

## 2.3 ATTENTION-GUIDED KV CACHE COMPRESSION

To address the accuracy degradation of block prefill eviction, prior work employs attention-based token scoring with a patched prompt. Here, token importance is quantified by the cross-attention it receives from query tokens: $\text{Attn}(x_t \to x_i)$ which denotes the attention weight from a query token $x_t$ to a key token $x_i$. Tokens that receive higher attention from queries are considered more important, while those with lower scores are evicted to satisfy the memory budget $M$.

To provide guidance for cache eviction, the *patched prompt* strategy (Kim et al., 2024) appends an auxiliary prompt of length $p$ after each block ending at token $n$. These queries $x_{n+1}, \ldots, x_{n+p}$ attend back to the preceding block tokens $x_1, \ldots, x_n$, as illustrated in Figure 1b. The resulting importance score $s_i$ of token $i$ is aggregated either by averaging or by taking the maximum, as defined in Equation (3). The patched prompt is used for scoring and not retained in the KV cache, and all experiments in this paper adopt the maximum aggregation.

$$s_i^{\text{avg}} = \frac{1}{p} \sum_{t=n+1}^{n+p} \text{Attn}(x_t \to x_i), \quad s_i^{\text{max}} = \max_{t \in [n+1, n+p]} \text{Attn}(x_t \to x_i). \tag{3}$$

In block prefill, accuracy is highly dependent on the content of the patched prompt. To study this, Figure 2 presents a controlled experiment where we assume oracle access to the future user query, with inserting it as the patched prompt yielding the highest accuracy (Exact-Question)[1]. Since the dialogue history $\mathcal{H}$ in Equation (1) consists of question-answer turns, it offers an opportunity to approximate the future query with semantically related turns. To test this idea, we embed both user queries $q_1, \ldots, q_{N_q}$ and conversation utterances $u_1, \ldots, u_{N_u}$ into a shared embedding space, compute semantic similarity scores, and construct patched prompts using the top-ranked multi-turns.

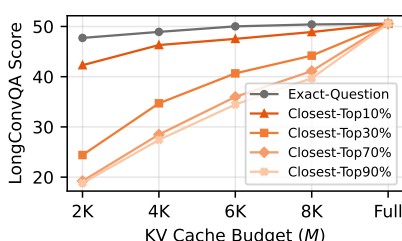

As expected, the results confirm that conversation utterances semantically aligned with the future question can serve as effective proxies, with the closest 10% achieving accuracy nearly matching the Exact-Question case in Figure 2. Accuracy then declines as the selected segments become less related, showing that the degree of semantic alignment directly determines the quality of answers generated from the compressed KV cache. These findings narrow our objective: during block prefill eviction, the central challenge is to identify a patched prompt that best approximates unseen questions without any supervision from future conversation. To this end, we employ unsupervised clustering methods to discover dialogue segments aligned with future queries, which will be introduced in Section 3.1.

Figure 2: **Patched-prompt analysis**: LoCoMo results with LLaMA3.1-8B under block prefill. Patched prompts are formed by selecting the top 10%–90% similar conversation utterances to $q_i$.

## 3 METHOD

### 3.1 EPISODIC KV CACHE MANAGEMENT WITH CONVERSATION CLUSTERING

Conversations can be segmented into coherent episodes (Galley et al., 2003; Sieber & Krenn, 2010), and subsequent utterances are naturally grounded in prior episodes. This episodic nature drives the core insight of EPICACHE: by clustering dialogue into episodes and constructing episode-specific caches, we can match an incoming query to the most relevant cache for accurate answer generation. As illustrated in Figure 3, this process unfolds in three stages—lightweight conversation clustering, episodic KV cache construction, and decoding with query matching.

**Stage 1. Conversation Clustering and Selecting Medoids.** For clustering conversation, we first divide the raw dialogue history $\mathcal{H}$ into segments of $w_{\text{embed}}$ utterances, denoted as $\mathcal{H} = S_1, \ldots, S_K$.

$$S_k = u_{(k-1)w_{\text{embed}}+1}, \ldots, u_{\min(kw_{\text{embed}}, N_u)}, \quad k = 1, \ldots, K, \quad K = \left\lceil \frac{N_u}{w_{\text{embed}}} \right\rceil \quad (4)$$

Each segment $S_k$ is encoded with a sentence encoders (Reimers & Gurevych, 2019) $f_{\text{embed}}$ into a vector embeddings $\mathbf{e}_k \in \mathbb{R}^d$, capturing the segment's semantics. We then apply K-Means clustering $\mathcal{C}(\cdot)$ to the embeddings $\{\mathbf{e}_k\}_{k=1}^K$, as illustrated in Figure 3a:

$$\mathcal{C}(\{\mathbf{e}_k\}_{k=1}^K) \to \{\mathcal{E}_1, \ldots, \mathcal{E}_E\}, \quad \bigcup_{e=1}^E \mathcal{E}_e = \{e_1, \ldots, e_K\}. \quad (5)$$

This procedure partitions $\mathcal{H}$ into $E$ semantically coherent topical episodes (Raedt et al., 2024). Examples of clustering results and episode-level utterance samples are shown in Figure A3.

For each cluster $\mathcal{E}_e$, we can compute its centroid embedding:

$$\mathbf{c}_e = \frac{1}{|\mathcal{E}_e|} \sum_{S_k \in \mathcal{E}_e} \mathbf{e}_k, \quad S_{\text{medoid}}^{(e)} = \arg \max_{S_k \in \mathcal{E}_e} \cos(\mathbf{e}_k, \mathbf{c}_e). \quad (6)$$

We then identify the *medoid* segment—i.e., the conversation segment in each cluster whose embedding is closest to the centroid in terms of semantic similarity. The medoid segment, as the representative of the cluster, contains multiple turns from both speakers and is used as the patched prompt in the subsequent block prefill eviction step.

---

[1]This strategy is infeasible in LongConvQA since (i) the future queries are unknown at compression time, and (ii) each new query requires re-prefill of entire context (Kim et al., 2025).

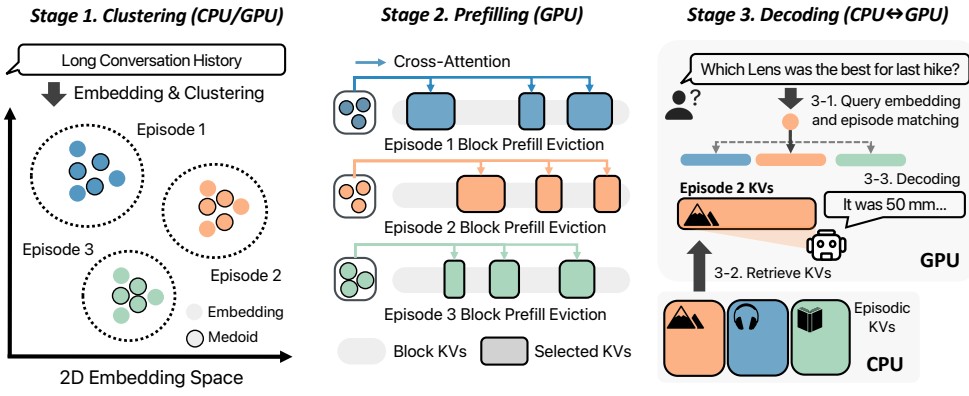

(a) Conversation Clustering     (b) Build Episodic KV Cache     (c) Decoding with Episodic KVs

Figure 3: **EpiCache Overview.** (a) offline segmentation and embedding of the conversation, followed by clustering into topical episodes. (b) Building episodic KV caches under a fixed GPU memory usage based on representative segments of each cluster. (c) an incoming query is embedded, matched to the closest episode, and the corresponding cache is retrieved for answer generation.

**Stage 2. Episodic KV Cache Compression.** As discussed in Section 2.3, patched prompts guide cache eviction toward retaining tokens relevant to the prompt content. Building on this insight, EPICACHE uses the medoid segment of each episode as the patched prompt, thereby collecting episode-specific KV entries under the memory budget.

For each episode $e \in \{1, \ldots, E\}$, we perform block prefill eviction over entire context, appending its patched prompt $\mathcal{P}_e$ after each block (Figure 3b). Attention scores are then computed with $\mathcal{P}_e$, and the top $M$ tokens are retained to form an episode-specific cache $C_{\mathrm{KV}}^{(e)}$. Finally, all episodic caches are collected into $\mathbb{B} = \{C_{\mathrm{KV}}^{(1)}, \ldots, C_{\mathrm{KV}}^{(E)}\}$ and stored offline for later retrieval.

**Stage 3. Query-KVs Matching and Decoding.** At decoding time, each user query $q_i$ is embedded with the same encoder $f_{\mathrm{embed}}$ used in clustering, ensuring that it lies in the same representation space as the episode centroids. The query is then matched to the closest centroid as follows:

$$\mathbf{q}_i = f_{\mathrm{embed}}(q_i), \quad e^{\dagger} = \arg \max_{e \in [1, E]} \cos(\mathbf{q}_i, \mathbf{c}_e). \tag{7}$$

As illustrated in Figure 3c, the framework retrieves the corresponding episodic cache $C_{\mathrm{KV}}^{(e^{\dagger})}$ from $\mathbb{B}$ and conditions generation on it: $f_{\mathrm{LM}}(q_i \mid C_{\mathrm{KV}}^{(e^{\dagger})}, )$. This design enables query-specific retrieval from episodic caches while keeping cache size bounded under memory budget $M$. This query-to-episode matching introduces overhead from embedding, matching, and retrieval; however, the cost is minor compared to decoding and is further examined in Section 4.4.

## 3.2 SENSITIVITY-AWARE LAYER-WISE KV BUDGET ALLOCATION

We further address the accuracy degradation of block prefill by proposing KV cache *budget allocation* strategy. The key idea is to measure how much each layer's Key state representation deviates under block prefill and to distribute a KV budget across layers in proportion to this deviation.

**Simulating Block Prefill via Custom Masking.** To quantify the deviation caused by block prefill eviction, we introduce a custom masking scheme. Each transformer layer is represented as a function $f$ that takes the previous layer's output $X^{(\ell-1)} \in \mathbb{R}^{N \times d}$. Here, $N$ denotes the sequence length and $d$ the hidden dimension. The function produces the $\ell_{\mathrm{th}}$ layer output $X^{\ell} = f\left(X^{\ell-1}, \mathcal{M}\right)$, where $\mathcal{M}$ is the standard causal mask.

We replace $\mathcal{M}$ with a custom mask $\mathcal{M}'$ that enforces a budget $M$, attending to sink tokens and the most recent tokens (Xiao et al., 2024). This design follows static compression methods, allowing us to simulate block prefill eviction in a single forward pass and directly measure its effect on layer representations.

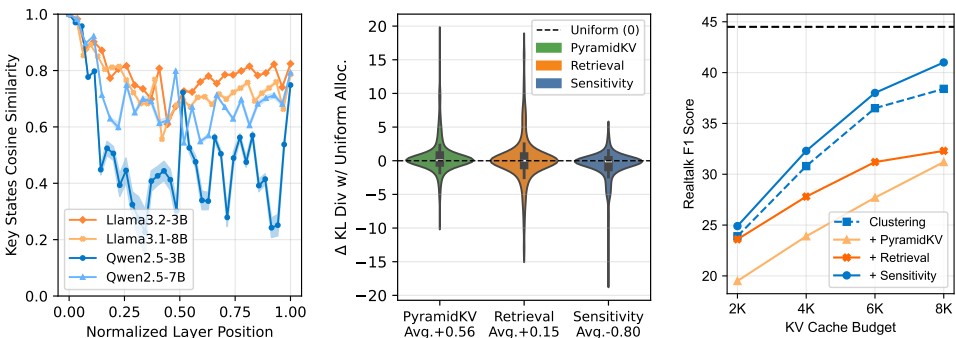

(a) Key Similarity across Layers   (b) $\Delta$ KL Divergence with Full KV   (c) LongConvQA Accuracy

Figure 4: **Layer-wise Sensitivity Analysis and KV Budget Allocation.** (a) Key states cosine similarity across normalized layer positions. (b) KL divergence is measured between block pre-fill ($M$=4K) and full KV answer predictions, with uniform allocation as the baseline. Per-sample KL divergence shifts are shown when applying three allocation strategies—sensitivity-aware, Pyra-midKV, and retrieval head profiling-on the Realtalk benchmark with Qwen2.5-7B. (c) Realtalk accuracy comparison across budget allocation methods using Qwen2.5-7B.

**Layer Sensitivity Guided KV Budget Allocation.** We quantify the per-layer impact of block prefill eviction by comparing Key states[2] produced under the causal mask $\mathcal{M}$ and the custom mask $\mathcal{M}'$. For each layer $\ell$, the forward pass under each mask produces:

$$K_{\text{full}}^{\ell} = f\left(X_{\text{full}}^{\ell-1}, \mathcal{M}\right) W_K^{\ell}, \qquad K_{\text{block}}^{\ell} = f\left(X_{\text{block}}^{\ell-1}, \mathcal{M}'\right) W_K^{\ell}, \tag{8}$$

where $K_{\text{full}}^{\ell}$ and $K_{\text{block}}^{\ell}$ are the $l$-th layer Key states computed under $\mathcal{M}$ and $\mathcal{M}'$, respectively. We then define layer sensitivity as the average cosine similarity between the two sets of Key vectors across attention heads and input tokens:

$$\sigma_{\ell} = \frac{1}{HN} \sum_{h=1}^{H} \sum_{i=1}^{N} \cos\left(k_{\text{full},i}^{(\ell,h)}, k_{\text{block},i}^{(\ell,h)}\right) \tag{9}$$

Empirically, $\sigma_{\ell}$ exhibits large variation across layers yet remains consistent across different inputs in Figure 4a (shadowed regions denote input variance), indicating that sensitivity is model-dependent rather than input-dependent. We define $s_{\ell} = 1 - \sigma_{\ell}$ as the sensitivity score for layer $\ell$.

Based on this observation, we propose a sensitivity-aware budget allocation strategy that assigns larger cache budgets to layers more sensitive to block prefill and smaller budgets to less sensitive ones. Specifically, we redistribute the global budget $(M \cdot L)$ according to layer sensitivity scores $s_{\ell}$, with $\alpha$ controlling how sharply the allocation emphasizes sensitive layers:

$$M_{\ell}^{\text{alloc}} = \frac{s_{\ell}^{\alpha}}{\sum_{j=1}^{L} s_j^{\alpha}} \cdot (L \cdot M), \qquad \sum_{\ell=1}^{L} M_{\ell}^{\text{alloc}} = L \cdot M, \tag{10}$$

We evaluate this approach by measuring how much budget allocation shifts the KL divergence between block prefill and full KV cache answer predictions, where negative values indicate closer alignment to full KV cache answer generation.

As shown in Figure 4b, sensitivity-aware allocation shifts KL divergence by –0.80 relative to uni-form allocation. In contrast, prior allocation strategies such as PyramidKV (Cai et al., 2025), which follows a pyramid-shaped budgeting, and retrieval head profiling based allocation (Wu et al., 2025b) tend to increase KL divergence under block prefill. This gap is directly reflected in task performance: as shown in Figure 4c, sensitivity-aware allocation consistently improves LongConvQA accuracy and complements episodic cache management framework, while other allocation strategies severely degrade accuracy.

---

[2]Further details regarding the rationale for using Key states deviation are provided in Appendix C.1.

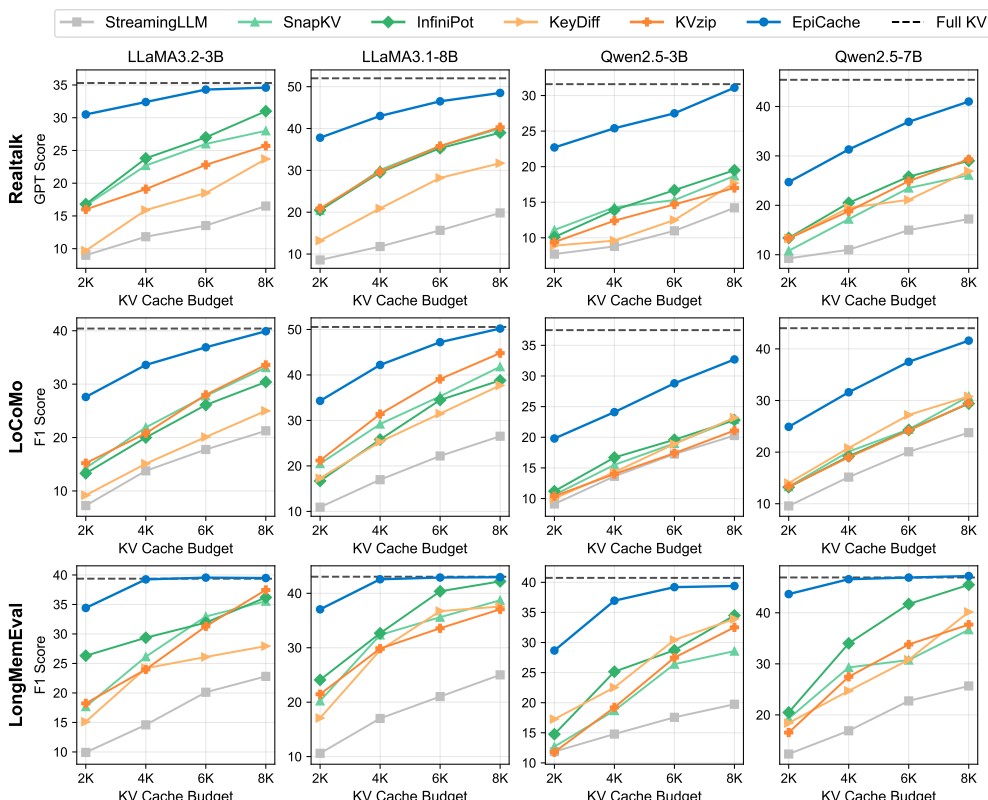

Figure 5: **LongConvQA Evaluation Results** (Realtalk, LoCoMo, and LongMemEval) results with fixed KV cache budget size-$M$ across four LLMs. The number of episodes (clusters) fixed to $E{=}4$ in all experiments. The average full KV lengths of the three benchmarks are 26K, 21K, and 20K.

## 4 EXPERIMENTS

### 4.1 SETUP

**Models and Benchmarks.** We evaluate on four pretrained LLMs: LLaMA-3.2-3B, LLaMA-3.1-8B (Grattafiori et al., 2024), Qwen2.5-3B, and Qwen2.5-7B (Qwen et al., 2025). All experiments follow the LongConvQA setup in Equation (2), where models answer various queries grounded in long conversation histories with compressed KV cache. We use three benchmarks: Realtalk (Lee et al., 2025) and LoCoMo (Maharana et al., 2024), containing multi-day human-human dialogues, and LongMemEval (Wu et al., 2025a), consisting of multi-session user-LLM conversations. Further details of LongConvQA benchmarks are provided in Appendix A.1.

**Baselines.** We compare against comprehensive KV cache compression methods adapted to block prefill. StreamingLLM (Xiao et al., 2024) applies static retention of sink and recent tokens, SnapKV, InfiniPot, and KVzip (Li et al., 2024; Kim et al., 2024; 2025) use patched prompt based attention scoring, and KeyDiff (Park et al., 2025) scores tokens based on similarity among Key states and retains those with distinctive representations. Detailed baseline setting can be found in Appendix A.2.

**EPICACHE Setup.** EPICACHE clusters the conversation history $\mathcal{H}$ into episodic segments and performs cache compression with patched prompts, while allocating per-layer budgets based on sensitivity measurements. We use $E{=}4$ episodes and Qwen3-0.6B (Zhang et al., 2025) as the embedding model. Layer sensitivities are profiled once on a BookSum (Kryściński et al., 2022) sample for each LLM and reused across all experiments, with the sharpness hyper-parameter set to $\alpha{=}1.1$ for the LLaMA series and $\alpha{=}1.3$ for the IQwen series. The overall process is detailed in Algorithm 1, while further setup specifications are provided in Appendix A.3.

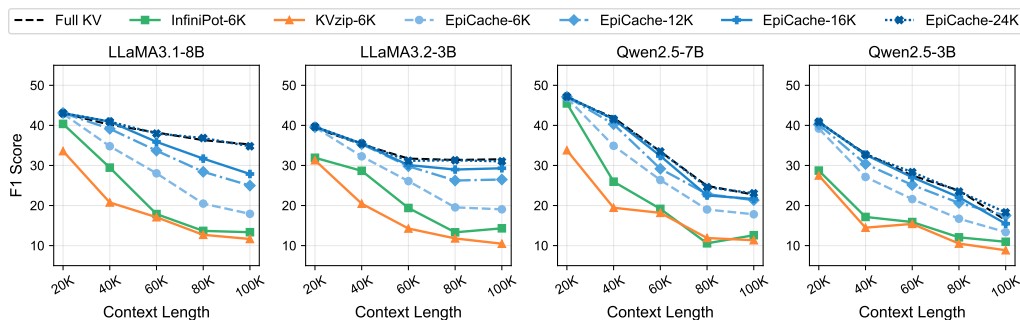

Figure 6: **Memory Scalability up to 100K Context.** Conversation histories between user and LLM-based assistant scaled to 100K tokens across four LLMs with LongMemEval. Comparison of InfiniPot and KVzip ($M$=6K) with EPICACHE (4 episodes, $M$=6K–24K).

### 4.2 MAIN EVALUATION RESULTS

**LongConvQA Evaluation.** Figure 5 shows results on the three LongConvQA bench-marks—Realtalk, LoCoMo, and LongMemEval—under block prefill eviction with varying cache budget $M$. StreamingLLM consistently shows the lowest accuracy, while patched-prompt methods (SnapKV, InfiniPot, KVzip) and Key states similarity-based KeyDiff suffer significant accuracy degradation compared to full KV. EPICACHE achieves consistently higher scores across all models and benchmakrs, exceeding baselines by up to 20 points under tightly compressed budgets ($M$=2–4K), while approaching full KV performance at budgets ($M$=6–8K). These consistent improvements across the three benchmarks highlight the effectiveness of episodic KV cache compression, along with sensitivity-aware budget allocation, in preserving dialogue history under fixed memory constraints. Detailed results including sub-task accuracy are provided in Appendix D.

**Memory Scalability Evaluation.** Figure 6 evaluates LongConvQA under extended conversation lengths, scaling up to 100K tokens.[3] Open-source LLMs exhibit declining QA performance as context length grows to 100K, as reported in Wu et al. (2025a), and the performance gap between full KV and baseline methods (KVzip, InfiniPot) becomes increasingly pronounced. EPICACHE delivers higher accuracy than baselines at the same memory budget across all context lengths, and as the KV cache budget increases, its accuracy steadily approaches full KV, demonstrating the memory scalability of our approach.

### 4.3 ABLATION STUDY

We conduct ablation studies to examine the design choices of EPICACHE. The detailed setups and results are provided in Appendix B, and we summarize the design aspects here:

- **Alternative design**: replacing episodic cache management with a *RAG-like* approach that directly inputs clustered conversation segments with the query.
- **EPICACHE ablation studies**: effect of block size $M_{block}$; window size $w_{embed}$; encoder choice $f_{embed}$; number of episodes $E$; number of medoids for patched prompts.

### 4.4 EFFICIENCY ANALYSIS

Figure 7a reports decoding latency breakdown and peak GPU memory usage, comparing full KV with EPICACHE under cache budgets of 2K-8K. Latency is evaluated per turn, consisting of answer decoding[4] and, for EPICACHE, query embedding, centroid matching, and KV cache retrieval. With fewer KV entries stored, EPICACHE reduces decoding latency by up to $2.4\times$ and peak GPU memory by $3.5\times$ compared to full KV. These results demonstrate that EPICACHE achieves both faster decoding and substantially lower memory usage, since block prefill constrains peak memory to the fixed cache budget during both prefill and decoding.

---

[3]LongMemEval (Wu et al., 2025a) supports stacking conversation sessions with associated QA pairs, allowing conversation histories to be constructed at custom lengths.

[4]For fair comparison, decoding latency is measured by generating up to 10 tokens per turn.

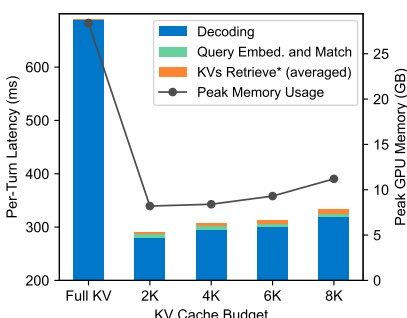 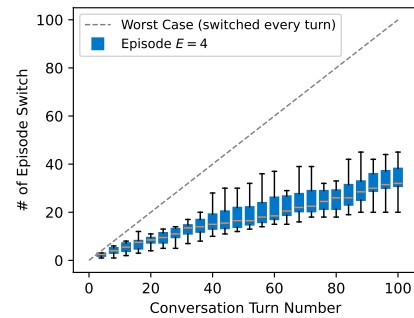

(a) Decoding latency breakdown and peak memory usage

(b) Number of episode switch in Realtalk

Figure 7: **Efficiency Analysis in Multi-Turn Conversation**: (a) Per-turn decoding latency and peak GPU memory for full KV (100K) and EPICACHE ($E$=4) with LLaMA-3.2-3B. Query Embed and Match: query encoding and centroid matching, KVs Retrieve: loading episodic cache from CPU to GPU memory. (b) Cumulative episode switches in Realtalk with $E$=4, showcasing how often episodes change across multi-turn conversation.

EPICACHE incurs retrieval overhead only when the conversation shifts topics, requiring a new episode cache to be loaded. To measure the frequency of such transitions, we cluster the initial portion of each conversation and embed subsequent turns to track their episode assignments. As shown in Figure 7b, cumulative transition counts grow sublinearly compared to the *Worst Case* line (episode switched every turn), indicating that topics often persist across turns. In such cases, the same episodic cache is reused without loading a new one, so no retrieval overhead occurs. As a result, the effective overhead remains under 5% in the per-turn latency breakdown of Figure 7a, while maintaining robust LongConvQA accuracy. Detailed measurements are provided in Appendix D.1.

## 5 RELATED WORK

**KV Cache Compression.** To mitigate the growing memory cost of KV caches, prior work either quantizes states into lower precision (Hooper et al., 2024; Liu et al., 2024b) or evicts less important tokens based on attention scores (Zhang et al., 2023; Li et al., 2024; Cai et al., 2025; Kim et al., 2025). As discussed in Section 2.2, most cache eviction methods follow the *post-prefill* approach, which requires full context prefilling and leads to unbounded peak memory. Block prefill methods such as InfiniPot (Kim et al., 2024), FINCH (Corallo & Papotti, 2024), and KeyDiff (Park et al., 2025) bound memory but suffer sharp accuracy drops in multi-turn conversation scenario.

**Retrieval-based Attention** Another line of work improves decoding efficiency by selectively retrieving only the most relevant parts of the KV cache for each query token, thereby reducing the cost of attention computation. Quest (Tang et al., 2024) retrieves KV entries at the granularity of pages, while SqueezedAttention (Hooper et al., 2025) and ClusterKV (Liu et al., 2024a) clusters Key states and loads the cluster most relevant to the query. A2ATS (He et al., 2025) apply vector quantization to construct codebooks and restore Key states to determine which parts of the KV cache to retrieve.

These methods share two key limitations. First, they operate in the *post-prefill* regime, assuming unbounded memory usage when building the retrieval index during prefill. Second, their retrieval units—pages, clusters, or codebooks—do not align with the episodic structure of conversations, limiting their applicability to LongConvQA under strict memory budgets.

## 6 CONCLUSION

EPICACHE is the first framework that combines block-wise prefill with episodic clustering and sensitivity-aware budget allocation to preserve topic-relevant context under a fixed memory budget. Across multiple LongConvQA benchmarks, EPICACHE substantially outperforms existing compression methods, demonstrating that efficient multi-turn interaction is feasible even under strict resource constraints and marking a practical step toward memory-efficient conversational AI.

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

# A EXPERIMENTAL DETAILS

## A.1 DATASET

We evaluate EPICACHE on three LongConvQA benchmarks: Realtalk (Lee et al., 2025), Lo-CoMo (Maharana et al., 2024), and LongMemEval (Wu et al., 2025a). Three benchmarks follow the LongConvQA formulation in Section 2.1, where a long conversational history $\mathcal{H}$ is provided and the model is required to answer a sequence of queries $\mathcal{Q} = q_1, \ldots, q_{N_q}$ grounded in dialogue history. This formulation evaluates the answer accuracy of LLMs in a *multi-turn conversation*.

**Realtalk.** Realtalk (Lee et al., 2025) is a real-world dataset of 10 long-term conversations, where pairs of participants engaged in daily messaging for 16-21 days. Unlike LLM-simulated corpora such as LoCoMo, Realtalk captures natural dialogue including typos, abbreviations, asynchronous response gaps, and consecutive messages, while also reflecting diverse emotional expressions and shifts in persona consistency.

For evaluation, the dataset provides annotated memory probing questions across three subtasks-multi-hop, temporal reasoning, and commonsense—requiring models to recall and reason over extended histories. Following the original setup, we adopt GPT-based scoring (gpt-4o-mini-2024-07-18) to assess open-ended generation.

**LoCoMo.** LoCoMo (Maharana et al., 2024) is a benchmark of long-term conversations, created through a human-machine pipeline where LLM-based agents generate dialogues grounded in distinct personas and temporal event graphs, and human annotators refine them for long-range consistency. The dataset consists of 10 conversations, each spanning up to 35 sessions with around 300 turns. The QA benchmark is divided into five subtasks: (i) Single-hop, (ii) Multi-hop, (iii) Temporal reasoning, (iv) Open-domain knowledge, and (v) Adversarial. Evaluation is conducted with open-ended genetation with F1 score.

We exclude the *adversarial* subtask for the following reason. This task tests whether a model can recognize unanswerable questions by choosing between a plausible but incorrect answer and a "no such information" response. Under KV cache compression, however, models frequently over-predict the latter, which leads to spuriously high scores. For example, with LLaMA-3.2-3B the adversarial score is only 12.11 under full KV, yet jumps to 49.78 with 4K KVzip compression—an increase that reflects bias rather than genuine improvement.

This behavior contrasts with other subtasks such as temporal or multi-hop reasoning, where compressed caches consistently degrade performance. Because open-source models already struggle on adversarial questions (Maharana et al., 2024), reporting these inflated numbers would give a misleading evaluation of answer quality under compressed KV cache. We therefore omit adversarial results from our main evaluation and defer a more thorough study of unanswerability detection under compression to future work.

**LongMemEval.** LongMemEval (Wu et al., 2025a) benchmarks long-term memory in user-assistant interactions with five core abilities—information extraction, multi-session reasoning, temporal reasoning, knowledge updates, and abstention—through seven question types (single-session user/assistant/preference, two-hop, multi-session synthesis, knowledge update, temporal reasoning, and abstention). A key property is its *length-configurable* chat histories: the benchmark provides standardized settings with extremely long contexts (e.g., up to 1.5M tokens), designed as controlled stress tests of memory and retrieval mechanisms. We follow the open-ended generation setup and report F1 scores for this dataset.

To align LongMemEval with the LongConvQA formulation in Section 2.1, we utilize the custom session stacking provided by LongMemEval[5] to build coherent long conversations from user-LLM. Using this feature, we construct evaluation sets while preserving the original distribution of all question types. Specifically, we sample QA pairs according to the benchmark's task-type proportions, retrieve the corresponding evidence conversation sessions, and assemble them into chronologically consistent histories. We then evaluate models at context lengths of 20K, 40K, 60K, 80K, and 100K tokens. This design allows us to test KV cache compression under scalable memory budgets.

---

[5]https://github.com/xiaowu0162/LongMemEval

## A.2 KV CACHE COMPRESSION BASELINE SETUP

We adapt existing KV cache compression methods to the block prefill setting for a fair comparison with our approach. The baselines include both static retention and attention-based eviction strategies, as well as similarity-based selection.

**StreamingLLM.** Following StreamingLLM (Xiao et al., 2024), we retain a fixed number of sink and recent tokens throughout block prefill. Specifically, we fix the number of sink tokens to 128 for all models, while the remaining budget $M - 128$ is assigned to the most recent tokens.

**SnapKV.** We adapt SnapKV (Li et al., 2024) to the block prefill setting, where future queries are not accessible in LongConvQA. Following the original design, we use the window tokens—in our case the last 64 tokens of each block—as the patched prompt, and then apply the scoring function in Equation (3). Tokens with the highest attention relevance to this patched prompt are retained.

**KVzip.** We adapt KVzip (Kim et al., 2025) to block prefill by treating the entire block of tokens as the patched prompt. At each block boundary, we append a repetition instruction (e.g., "Repeat the part of the previous context exactly") followed by the full block tokens, and then apply the patched-prompt scoring method.

**InfiniPot.** We adopt the InfiniPot (Kim et al., 2024) by employing a general-purpose patched prompt designed to highlight globally important content. Specifically, we append the instruction "Summarize the previous context highlighting the most important parts." at the end of each block and compute scores according to Equation (3). This encourages selection of semantically informative tokens across the block.

**KeyDiff.** KeyDiff (Park et al., 2025) is KV cache eviction method for block prefill. For each block, it constructs an anchor key by averaging the key states of all tokens, and computes the dissimilarity score of each token as the negative cosine similarity between its key state and the anchor. These scores are then used to guide eviction. Following the original implementation, we evaluate $M_{\text{block}} \in 128, 512, 1024, 2048$, which includes the default setting of 128, and report results using the configuration that achieved the best performance.

To ensure fairness, all attention-based methods (SnapKV, InfiniPot, KVzip, EPICACHE) use the same scoring formulation from Equation (3), and all eviction methods are combined with head-wise non-uniform token selection as suggested by Feng et al. (2024). In practice, we adopt the max aggregation for $s_i^{\max}$ rather than averaging across heads, since empirical results consistently showed superior performance across all baselines.

## A.3 EPICACHE SETUP

**Overall Process.** We provide the complete procedure of EPICACHE in Algorithm 1. The framework consists of two phases. In **Phase A**, the conversation history is clustered into topical episodes and a compressed KV cache is prepared for each episode. In **Phase B**, online queries are answered by retrieving the most relevant episodic cache.

In *Phase A1*, we segment the conversation history, embed each segment, and cluster them into $E$ episodes. This step can be performed offline, and the cost of segment encoding and K-means clustering is negligible (under a minute). Each episode is represented by its centroid and a medoid segment that serves as a patched prompt. In *Phase A2*, we measure per-layer sensitivity by comparing Key states under full and block-prefill masks, and allocate layer-wise budgets proportionally using the sharpness hyper-parameter $\alpha$. In *Phase A3*, we construct episodic KV caches by performing block-wise prefill with the patched prompt appended, and then compress the resulting caches according to the allocated budgets from Phase A2. Although prefill must be repeated for every episode, the peak memory remains flat during this process (see Figure 1c), making it practical for constrained-memory environments.

In Phase B, when a new query arrives, we embed the query and compute its similarity to the episode centroids. The query is then routed to the most relevant episodic cache, which is loaded for decoding.

---

**Algorithm 1** EPICACHE with Layer-wise Budget Allocation Pseudo Code

---

**Require:** $\mathcal{H}$ (history, $N_u$ turns), $f_{\text{embed}}$, $w_{\text{embed}}$, $E$, $M$; $f_{\text{LM}}$ ($L$ layers, $H$ heads); masks $\{\mathcal{M}, \mathcal{M}'\}$; sharpness $\alpha$; calibration batch $\mathcal{B}$ with $|\mathcal{B}| = 1$; Patched prompt $\mathcal{P}_e$ (built from medoid segments, see stage A1)

**Ensure:** Episodic caches $\mathbb{B} = \{C_{\text{KV}}^{(1)}, \ldots, C_{\text{KV}}^{(E)}\}$, centroids $\{\mathbf{c}_e\}_{e=1}^E$, and layer budgets $\{M_\ell^{\text{alloc}}\}_{\ell=1}^L$

    **Phase A: Clustering and Prefill**

    *A1. Conversation Segment & Clustering (Offline)*

1: Partition $\mathcal{H}$ into $K = \lceil N_u / w_{\text{embed}} \rceil$ segments $\{S_k\}_{k=1}^K$ and encode $\mathbf{e}_k = f_{\text{embed}}(S_k)$.

2: Run K-Means Clustering on $\{\mathbf{e}_k\}$ to obtain $\{\mathcal{E}_e\}_{e=1}^E$.

3: **for** $e = 1$ **to** $E$ **do**

4:     $\mathbf{c}_e \leftarrow \frac{1}{|\mathcal{E}_e|} \sum_{S_k \in \mathcal{E}_e} \mathbf{e}_k$;   $S_{\text{medoid}}^{(e)} \leftarrow \arg\max_{S_k \in \mathcal{E}_e} \cos(\mathbf{e}_k, \mathbf{c}_e)$

5:     Build patched prompt $\mathcal{P}_e$ by concatenating utterances of $S_{\text{medoid}}^{(e)}$.

6: **end for**

    *A2. Measure layer sensitivity & allocate KV budgets.*

7: **for** each $x \in \mathcal{B}$ **do**

8:     $K_{\text{full}}^\ell(x) \leftarrow f_{\text{LM}}(x, \mathcal{M}) W_K^\ell$;   $K_{\text{block}}^\ell(x) \leftarrow f_{\text{LM}}(x, \mathcal{M}') W_K^\ell$   for $\ell = 1$:$L$

9:     $\sigma_\ell(x) \leftarrow \frac{1}{HN} \sum_{h=1}^H \sum_{i=1}^N \cos\big(k_{\text{full},i}^{(\ell,h)}(x), k_{\text{block},i}^{(\ell,h)}(x)\big)$

10: **end for**

11: $\sigma_\ell \leftarrow \frac{1}{|\mathcal{B}|} \sum_{x \in \mathcal{B}} \sigma_\ell(x)$;   $s_\ell \leftarrow 1 - \sigma_\ell$

12: $w_\ell \leftarrow \dfrac{s_\ell^\alpha}{\sum_{j=1}^L s_j^\alpha}$;   $M_\ell^{\text{alloc}} \leftarrow (L \cdot M)\, w_\ell$

    *A3. Build episodic KV caches.*

13: **for** $e = 1$ **to** $E$ **do**

14:     Block-wise prefill over $\mathcal{H}$, appending $\mathcal{P}_e$ to each block of $M_{\text{block}}$ tokens.

15:     Compute scores w.r.t. $\mathcal{P}_e$ with Equation (3) and retain the top $M$ tokens.

16:     $C_{\text{KV}}^{(e)} \leftarrow$ compressed cache for episode $e$.

17: **end for**

18: $\mathbb{B} \leftarrow \{C_{\text{KV}}^{(1)}, \ldots, C_{\text{KV}}^{(E)}\}$.

    **Phase B: Online decoding**

19: For query $q_i$: $\mathbf{q}_i \leftarrow f_{\text{embed}}(q_i)$;   $e^\dagger \leftarrow \arg\max_e \cos(\mathbf{q}_i, \mathbf{c}_e)$

20: Retrieve $C_{\text{KV}}^{(e^\dagger)}$ and generate with compressed cache: $f_{\text{LM}}(q_i \mid C_{\text{KV}}^{(e^\dagger)})$.

---

If the same cache is selected as in the previous turn, no additional retrieval is required since the cache remains resident, further reducing overhead.

**Detailed Settings** For segment construction, we set the embedding window size $w_{\text{embed}}$ to 4, selected from $2, 4, 8$. To cluster segments into episodes, we apply the standard K-means algorithm with k-means++ initialization (Arthur & Vassilvitskii, 2007). This offline segmentation and clustering stage completes within one minute, incurring negligible overhead.

For sensitivity-aware budget allocation, we estimate per-layer weights using a single randomly sampled long document from the BookSum (Kryściński et al., 2022) dataset, chosen to avoid bias from any specific conversational dataset. By performing two forward passes—one with the full causal mask and one with the block-prefill mask—we measure layer-wise deviations and compute allocation weights. Because only one sample is used, the overhead of this calibration step is negligible.

In block prefill, the cache always maintains size $M$: as conversation segments are added in blocks of $M_{\text{block}}$, the cache can temporarily grow up to $M + M_{\text{block}}$ entries, after which eviction is applied to reduce it back to $M$. A larger $M_{\text{block}}$ enables the model to cover the entire conversation more quickly but increases the temporary peak memory footprint, while a smaller $M_{\text{block}}$ lowers peak memory at the cost of slower coverage. We set $M_{\text{block}} \in 128, 512, 1024, 2048$ to balance this trade-off.

For the patched prompt, we use the medoid segment of size 8, selected from $4, 8, 12$. Ablation studies show that different segment sizes yield only marginal performance differences. After constructing episodic KV caches, we offload them to offline memory (e.g., CPU) to minimize GPU memory usage. During online decoding, episodic caches are retrieved from offline memory. This design enables constrained GPU memory usage while keeping retrieval overhead manageable. A comparison of this design choice is provided in the following section.

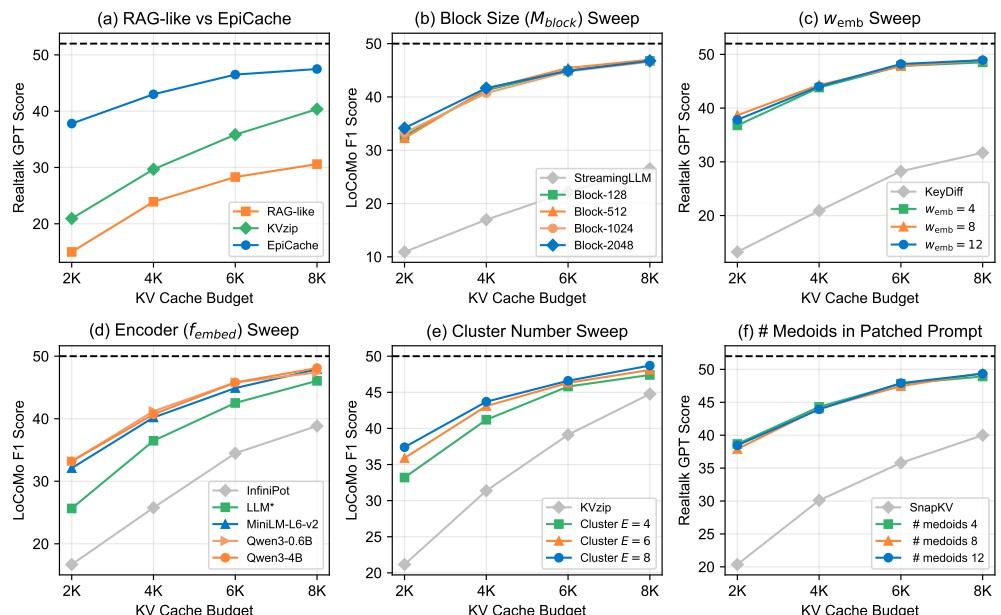

Figure A1: **Alternative Design Exploration**: (a) Comparison with *RAG-like* baseline, which feeds clustered segments directly with query-based matching. (b) Sweep of block prefill size $M_{\text{block}}$. (c) Effect of segmentation window size $w_{\text{emb}}$. (d) Encoder choice for clustering and query embedding. (e) Effect of episode number $E$ in episodic clustering. (f) Number of medoid samples in patched prompts. All experiments use LLaMA3.1-8B on the Realtalk and LoCoMo benchmarks.

## B    ALTERNATIVE DESIGN EXPLORATION

### B.1    COMPARISON WITH RAG-LIKE APPROACH.

A representative alternative design is a *RAG-like* approach, where conversation segments are clustered and the query is used to select the most relevant clusters, which are then directly fed into the LLM to form a KV cache under a fixed budget. The key difference from block prefill eviction is that block prefill spans the entire dialogue and selects cluster-related tokens, whereas the RAG-like method isolates only the chosen clusters when constructing the KV cache. As shown in Figure A1 (a), this leads to substantially worse performance—not only below EPICACHE but also below block prefill-based compression methods such as KVzip.

This result further suggests that clustering alone is insufficient for directly constructing episodic memory inputs: conversation histories segmented only by clustering remain under-contextualized. Developing more effective clustering or episode partitioning strategies tailored to RAG-like design thus remains an interesting direction for future work.

### B.2    EPICACHE ABLATION STUDY.

**Block Size.**    Figure A1 (b) varies the block size $M_{\text{block}}$ used during block prefill. Accuracy differences remain minor across settings. In practice, smaller block sizes reduce peak memory but require more iterations to span the full context, whereas larger blocks incur higher memory during prefill but process the history more efficiently. This reflects a memory-latency trade-off, and block size can be tuned according to deployment constraints.

**Clustering Design.**    Figures Figure A1 (c-f) examine clustering-related hyper-parameters. (i) Figure A1 (c) sweeps the segmentation window size $w_{\text{emb}}$ for utterance grouping, showing little sensitivity and indicating robustness to segmentation granularity. (ii) Figure A1 (d) varies the encoder for clustering and query embedding. The LLM's own embedding layer yields weaker performance, while lightweight sentence encoders (MiniLM-L6-v2, 0.06B; Qwen3-0.6B) deliver strong gains

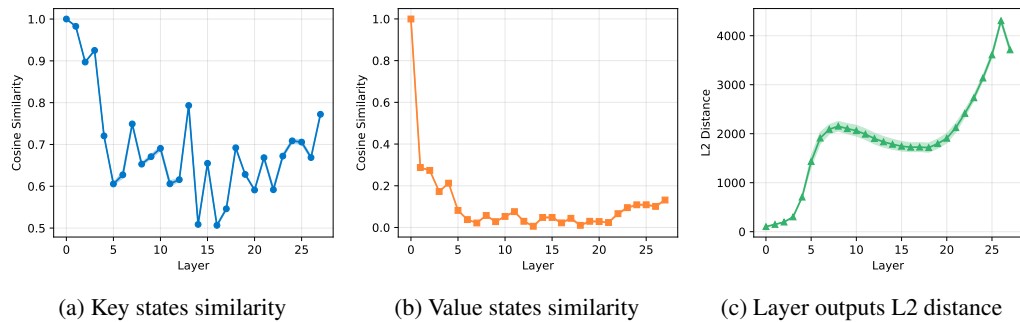

(a) Key states similarity       (b) Value states similarity       (c) Layer outputs L2 distance

Figure A2: **Layer-wise Sensitivity Analysis.** layer-wise deviation results under the full and block masks using Qwen2.5-7B on LoCoMo conversation history. Key and Value states are measured by cosine similarity, while hidden states at layer outputs are measured by L2 distance; shaded regions indicate variance across input samples.

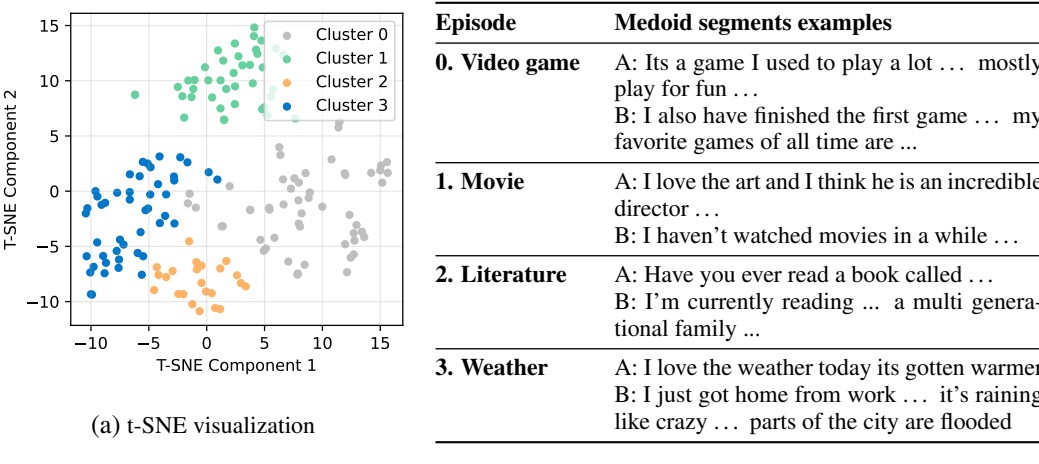

(a) t-SNE visualization

(b) Medoid samples by cluster.

Figure A3: **Episodic clustering of conversation segments.** (a) t-SNE visualization of conversation clustering. (Silhouette score=0.28) (b) Medoid segments illustrate coherent topics per cluster.

over InfiniPot. Larger encoders (Qwen3-4B) add only marginal improvements, suggesting that lightweight sentence encoders are sufficient for conversation clustering in EPICACHE. (iii) Figure A1 (e) sweeps the number of episodes $E$, where larger $E$ yields finer-grained topic segmentation and more accurate episodic caches—especially under tight budgets—at the cost of maintaining more caches offline. (iv) Figure A1 (f) varies the number of medoid samples in patched prompts, with little effect, suggesting that only a few representatives per cluster are sufficient.

## C  FURTHER ANALYSIS

### C.1  BLOCK-PREFILL SENSITIVITY ANALYSIS

We analyze layer-wise deviations under block prefill by comparing multiple internal states—Key, Value, and layer outputs—across Transformer layers computed with the full causal mask ($\mathcal{M}$) and the block mask ($\mathcal{M}'$). To this end, we forward LoCoMo (Maharana et al., 2024) conversation history samples under both masks and plot the resulting internal states differences across layers, as defined in Equations (8) and (9). Specifically, each plot reports cosine similarity of Key and Value states, and L2 distance of layer outputs, respectively.

We find that Value states (Figure A2b) exhibit consistently low similarity across layers, offering little discriminative trend. Layer outputs, measured by L2 distance (Figure A2c), show a monotonic error

| Method | $M$ | LLaMA3.2-3B | | | | | Qwen2.5-3B | | | | |
|--------|-----|-------------|------|------|------|------|------------|------|------|------|------|
| | | Decode (ms) | Embed. (ms) | Retr. (ms) | Peak Mem. (GB) | KVs (GB) | Decode (ms) | Embed. (ms) | Retr. (ms) | Peak Mem. (GB) | KVs (GB) |
| Full KV | 100K | 68.9 | – | – | 28.4 | 11.6 | 54.1 | – | – | 20.7 | 3.8 |
| EPICACHE | 2K | 28.1 | 6.1 | 2.0 | 8.2 | 0.2 | 36.7 | 5.2 | 1.2 | 7.6 | 0.1 |
| | 4K | 29.6 | 6.0 | 4.0 | 8.4 | 0.5 | 37.7 | 5.5 | 2.2 | 8.8 | 0.2 |
| | 6K | 30.1 | 5.5 | 6.0 | 9.3 | 0.7 | 38.9 | 5.4 | 2.9 | 9.5 | 0.2 |
| | 8K | 32.2 | 5.6 | 7.9 | 11.2 | 0.9 | 42.3 | 5.6 | 3.7 | 10.5 | 0.3 |

Table A1: Runtime and memory comparison under full KV and block prefill across LLaMA and Qwen. Columns: $M$ (memory budget), Decode (ms), Embed. (ms), Retrieve. (ms), Peak Memory (GB), and KVs Storage (GB). Latency is averaged per turn over 100 multi-turn queries on Long-MemEval with 100K context length, reported across five runs. Measurements are conducted on an NVIDIA DGX A100 system.

accumulation pattern rather than meaningful variation. In contrast, Key states (Figure A2a) provide clear differentiation across layers. This observation motivates our use of Key state deviation as the sensitivity measure for budget allocation, as discussed in Section 3.2. Further analysis of why these trends differ across Key, Value, and layer-wise output representations is left for future work.

### C.2 CONVERSATION CLUSTERING ANALYSIS

In this section, we provide qualitative examples of conversation clustering to illustrate how episodic structures emerge in practice. Conversation histories are divided into segments of $w_{embed} = 4$ utterances, which are then embedded using Qwen3-0.6B (Zhang et al., 2025). Segment embeddings are clustered with K-Means, and the resulting clusters are visualized in two dimensions via t-SNE, as shown in Figure A3 (a).

For each cluster, we further present representative medoid segments in Figure A3 (b). These examples demonstrate that the clustering procedure consistently groups segments into coherent topical episodes, such as games, movies, literature, or weather. The medoid samples highlight the interpretability of each episode and indicate how such episode-level partitioning can serve as the basis for episodic KV cache compression.

## D DETAILED EXPERIMENTAL RESULTS

### D.1 EFFICIENCY RESULTS

**Measurement Setup.** We use NVIDIA A100-40GB (PCIe) GPUs with dual Intel Xeon Platinum 8275CL CPUs. Latency is measured on LongMemEval (Wu et al., 2025a) with 100K context length, where each evaluation spans 100 multi-turn queries. We report the average per-turn latency across five runs for both LLaMA3.2-3B and Qwen2.5-3B.

**Results.** Table A1 presents the runtime and memory breakdown. Full KV incurs high decoding latency and large peak GPU memory, with LLaMA reaching 68.9 ms per turn and 28.4 GB memory. The KV cache storage of LLaMA (11.6 GB) also exceeds the model's parameter size. In comparison, Qwen2.5-3B shows a smaller KV cache size (3.8 GB), which can be attributed to its lower number of KV attention heads (2 vs. 8 in LLaMA) (Ainslie et al., 2023).

With EPICACHE, both models achieve substantial efficiency gains. On LLaMA3.2-3B, decoding latency drops from 68.9ms to 28.1ms (2.4× faster) and peak memory from 28.4GB to 8.2GB (3.5× smaller). For Qwen2.5-3B, latency improves from 54.1ms to 36.7ms (1.5× faster) and memory from 20.7GB to 7.6GB (2.7× smaller). The additional cost from query embedding and cache retrieval accounts for only about 5% of per-turn latency on average, confirming that the overhead introduced by episodic caching is minimal compared to the overall efficiency benefits.

## D.2 LONGCONVQA SUBTASK RESULTS

We provide detailed subtask results corresponding to the main LongConvQA experiments highlighted in Figure 5. For Realtalk and LoCoMo, results are reported in Table A2 for LLaMA3.2-3B and LLaMA3.1-8B and in Table A3 for Qwen2.5-3B and 7B. For LongMemEval, results across all four models are reported in Table A4.

## E FUTURE WORK

EPICACHE demonstrates that memory-bounded episodic KV caching can substantially improve accuracy on LongConvQA while maintaining minimal latency overhead. By clustering dialogue into episodes and dynamically retrieving episode-specific caches, it provides a practical solution that balances efficiency and effectiveness in resource-constrained environments.

There remain several promising directions for future research. First, while we adopt standard embedding-based clustering for episode construction, more advanced clustering strategies specialized for conversational structure (Raedt et al., 2024) can be developed. Such methods, orthogonal to our framework, can yield more coherent episodic boundaries and further strengthen the effectiveness of episodic KV cache management.

Second, EPICACHE currently operates with a fixed number of episodes. Extending it to adaptively determine the optimal number of episodes from the conversation history could improve scalability across diverse dialogue lengths and domains. In addition, incorporating KV cache quantization (Hooper et al., 2024; Liu et al., 2024b) into episodic caches would reduce cache storage and transfer costs, alleviating the movement overhead during the retrieval stage. Exploring these extensions would further enhance the practicality of episodic KV caching for long conversational scenario.

## F USE OF LARGE LANGUAGE MODEL

In accordance with the ICLR 2026 policy on large language model (LLM) usage, we disclose that we used an LLM solely to aid with grammar correction and minor phrasing improvements during the writing of this paper. All research ideas, technical contributions, and experiments were conceived, implemented, and validated entirely by the authors.

| LLaMA3.2-3B | | LoCoMo (Full KV length: 21.8K) | | | | | Realtalk (Full KV length: 26.4K) | | | |
|---|---|---|---|---|---|---|---|---|---|---|
| Method | $M$ | Multi-hop | Temporal | Open-domain | Single-hop | Avg | Multi-hop | Temporal | Common | Avg |
| Full KV | – | 36.0 | 15.1 | 13.2 | 54.5 | 40.3 | 39.0 | 30.8 | 38.2 | 35.3 |
| SnapKV (Li et al., 2024) | 2K | 17.3 | 3.7 | 17.3 | 17.6 | 14.3 | 20.1 | 10.5 | 25.0 | 16.6 |
| | 4K | 23.1 | 6.0 | 11.5 | 28.7 | 21.9 | 28.5 | 14.7 | 30.0 | 22.7 |
| | 6K | 27.3 | 9.9 | 10.6 | 36.8 | 27.8 | 31.7 | 18.2 | 33.2 | 26.0 |
| | 8K | 31.2 | 11.9 | 11.7 | 44.3 | 33.1 | 34.1 | 20.3 | 33.7 | 28.0 |
| InfiniPot (Kim et al., 2024) | 2K | 15.9 | 7.2 | 10.1 | 15.0 | 13.3 | 22.1 | 9.4 | 24.3 | 16.8 |
| | 4K | 21.3 | 12.0 | 10.0 | 23.7 | 20.0 | 29.2 | 16.0 | 31.9 | 23.8 |
| | 6K | 25.6 | 17.0 | 11.9 | 31.3 | 26.1 | 32.8 | 19.7 | 32.4 | 27.0 |
| | 8K | 28.6 | 20.5 | 11.4 | 36.9 | 30.4 | 35.6 | 24.5 | 36.9 | 31.0 |
| KeyDiff (Park et al., 2025) | 2K | 10.7 | 3.7 | 11.6 | 11.2 | 9.2 | 10.4 | 5.6 | 19.5 | 9.7 |
| | 4K | 15.7 | 8.3 | 11.5 | 17.9 | 15.1 | 16.8 | 14.2 | 18.2 | 15.9 |
| | 6K | 20.7 | 11.9 | 13.4 | 23.8 | 20.1 | 20.6 | 14.5 | 24.8 | 18.5 |
| | 8K | 23.7 | 15.7 | 14.7 | 30.2 | 25.0 | 24.8 | 20.7 | 29.7 | 23.7 |
| KVzip (Kim et al., 2025) | 2K | 22.0 | 4.5 | 12.0 | 17.3 | 15.2 | 18.7 | 9.5 | 27.9 | 16.0 |
| | 4K | 21.9 | 13.3 | 10.0 | 24.5 | 20.8 | 25.0 | 10.1 | 29.4 | 19.1 |
| | 6K | 28.1 | 16.5 | 12.0 | 34.2 | 28.0 | 26.1 | 15.2 | 36.3 | 22.8 |
| | 8K | 31.2 | 21.8 | 11.8 | 41.5 | 33.6 | 30.7 | 18.1 | 34.5 | 25.7 |
| EPICACHE | 2K | 29.3 | 15.9 | 13.7 | 33.1 | 27.6 | 34.7 | 23.0 | 41.3 | 30.5 |
| | 4K | 30.4 | 19.7 | 14.2 | 42.3 | 33.6 | 37.4 | 24.7 | 41.0 | 32.4 |
| | 6K | 33.7 | 22.2 | 12.1 | 46.4 | 36.9 | 39.9 | 25.6 | 44.2 | 34.3 |
| | 8K | 33.9 | 23.9 | 12.7 | 48.2 | 38.3 | 40.0 | 26.6 | 42.9 | 34.6 |

| LLaMA3.1-8B | | LoCoMo (Full KV length: 21.8K) | | | | | Realtalk (Full KV length: 26.4K) | | | |
|---|---|---|---|---|---|---|---|---|---|---|
| Method | $M$ | Multi-hop | Temporal | Open-domain | Single-hop | Avg | Multi-hop | Temporal | Common | Avg |
| Full KV | – | 43.1 | 22.7 | 17.2 | 67.4 | 50.5 | 49.2 | 55.9 | 48.4 | 52.0 |
| SnapKV (Li et al., 2024) | 2K | 23.4 | 6.6 | 14.3 | 25.5 | 20.5 | 20.2 | 15.8 | 34.3 | 20.4 |
| | 4K | 30.8 | 10.9 | 13.6 | 37.3 | 29.2 | 28.7 | 26.6 | 44.4 | 30.1 |
| | 6K | 33.3 | 13.8 | 13.3 | 46.8 | 35.3 | 37.3 | 32.2 | 42.3 | 35.8 |
| | 8K | 37.7 | 18.1 | 14.2 | 55.5 | 41.8 | 41.5 | 36.8 | 45.2 | 40.0 |
| InfiniPot (Kim et al., 2024) | 2K | 15.2 | 12.3 | 10.2 | 19.6 | 16.7 | 21.2 | 14.9 | 35.0 | 20.5 |
| | 4K | 23.0 | 22.1 | 13.9 | 29.5 | 25.8 | 30.0 | 25.8 | 39.1 | 29.5 |
| | 6K | 29.8 | 29.6 | 13.9 | 40.1 | 34.5 | 34.4 | 33.3 | 43.4 | 35.3 |
| | 8K | 33.1 | 33.8 | 13.8 | 45.5 | 38.8 | 38.3 | 38.2 | 43.4 | 39.0 |
| KeyDiff (Park et al., 2025) | 2K | 18.7 | 4.4 | 17.7 | 21.8 | 17.3 | 13.0 | 8.2 | 28.8 | 13.2 |
| | 4K | 23.3 | 14.1 | 13.8 | 31.5 | 25.3 | 21.9 | 16.5 | 30.9 | 20.9 |
| | 6K | 27.5 | 18.6 | 14.5 | 39.7 | 31.5 | 29.6 | 25.3 | 33.3 | 28.2 |
| | 8K | 33.0 | 25.8 | 16.4 | 46.2 | 37.7 | 33.0 | 29.9 | 33.3 | 31.7 |
| KVzip (Kim et al., 2025) | 2K | 24.4 | 8.4 | 22.8 | 24.7 | 21.2 | 20.6 | 16.6 | 34.6 | 20.9 |
| | 4K | 28.8 | 24.3 | 15.3 | 36.8 | 31.4 | 29.5 | 26.4 | 40.0 | 29.7 |
| | 6K | 32.8 | 31.0 | 13.3 | 47.3 | 39.1 | 36.4 | 33.4 | 41.5 | 35.8 |
| | 8K | 37.2 | 35.0 | 13.1 | 54.6 | 44.8 | 38.8 | 41.3 | 41.1 | 40.3 |
| EPICACHE | 2K | 33.2 | 26.3 | 14.2 | 43.5 | 36.3 | 37.9 | 35.2 | 45.2 | 37.8 |
| | 4K | 37.7 | 33.8 | 15.9 | 55.8 | 45.4 | 39.6 | 44.9 | 46.7 | 43.0 |
| | 6K | 38.2 | 36.4 | 16.4 | 58.2 | 47.3 | 42.3 | 50.5 | 35.2 | 46.5 |
| | 8K | 38.4 | 37.5 | 17.4 | 62.7 | 50.2 | 44.2 | 50.9 | 46.5 | 47.5 |

Table A2: **LongConvQA (LoCoMo and Realtalk) Evaluation**: Comparison of different KV cache compression methods under block-prefill with LLaMA series models.

| Qwen2.5-3B | | LoCoMo (Full KV length: 21.9K) | | | | | Realtalk (Full KV length: 26.6K) | | | |
|---|---|---|---|---|---|---|---|---|---|---|
| Method | $M$ | Multi-hop | Temporal | Open-domain | Single-hop | Avg | Multi-hop | Temporal | Common | Avg |
| Full KV | – | 33.2 | 22.9 | 12.3 | 49.1 | 38.4 | 32.7 | 28.0 | 39.6 | 31.6 |
| SnapKV (Li et al., 2024) | 2K | 14.1 | 8.0 | 11.5 | 10.4 | 10.6 | 12.4 | 5.6 | 23.5 | 11.1 |
| | 4K | 17.9 | 12.7 | 11.5 | 16.2 | 15.5 | 15.4 | 9.0 | 27.0 | 14.3 |
| | 6K | 21.3 | 13.1 | 13.7 | 21.1 | 19.0 | 17.9 | 9.3 | 26.2 | 15.3 |
| | 8K | 23.2 | 14.3 | 12.1 | 27.4 | 22.9 | 21.6 | 12.6 | 28.8 | 18.7 |
| InfiniPot (Kim et al., 2024) | 2K | 12.4 | 16.1 | 12.6 | 8.8 | 11.2 | 8.6 | 6.1 | 26.3 | 10.1 |
| | 4K | 18.2 | 19.7 | 10.2 | 15.7 | 16.7 | 15.4 | 8.9 | 24.3 | 13.9 |
| | 6K | 20.0 | 20.0 | 13.9 | 19.9 | 19.6 | 17.5 | 12.0 | 28.2 | 16.7 |
| | 8K | 23.9 | 18.8 | 13.2 | 25.1 | 22.8 | 23.9 | 11.4 | 31.1 | 19.5 |
| KeyDiff (Park et al., 2025) | 2K | 10.4 | 14.2 | 13.6 | 7.8 | 10.0 | 3.9 | 9.0 | 23.5 | 8.9 |
| | 4K | 14.0 | 15.8 | 12.0 | 14.1 | 14.3 | 9.2 | 6.0 | 21.6 | 9.6 |
| | 6K | 19.6 | 16.4 | 9.2 | 20.8 | 18.9 | 11.5 | 10.0 | 22.6 | 12.5 |
| | 8K | 21.6 | 17.3 | 7.4 | 27.8 | 23.2 | 18.1 | 15.1 | 24.4 | 17.7 |
| KVzip (Kim et al., 2025) | 2K | 11.8 | 6.1 | 11.9 | 11.4 | 10.4 | 10.6 | 5.3 | 18.2 | 9.4 |
| | 4K | 16.2 | 10.3 | 12.6 | 14.7 | 14.0 | 13.3 | 8.4 | 21.7 | 12.4 |
| | 6K | 19.5 | 12.7 | 11.3 | 19.1 | 17.4 | 15.0 | 10.0 | 27.8 | 14.7 |
| | 8K | 21.9 | 14.3 | 14.1 | 24.3 | 21.1 | 18.8 | 11.4 | 28.4 | 17.0 |
| EPICACHE | 2K | 23.6 | 7.6 | 13.4 | 23.8 | 19.8 | 25.6 | 13.2 | 37.7 | 22.0 |
| | 4K | 27.1 | 10.7 | 11.4 | 29.7 | 24.1 | 30.0 | 17.1 | 36.9 | 25.4 |
| | 6K | 27.3 | 13.3 | 10.8 | 37.2 | 28.8 | 32.6 | 19.7 | 36.4 | 27.5 |
| | 8K | 31.3 | 17.0 | 10.0 | 41.7 | 32.7 | 33.9 | 25.0 | 41.1 | 31.1 |

| Qwen2.5-7B | | LoCoMo (Full KV length: 21.9K) | | | | | Realtalk (Full KV length: 26.6K) | | | |
|---|---|---|---|---|---|---|---|---|---|---|
| Method | $M$ | Multi-hop | Temporal | Open-domain | Single-hop | Avg | Multi-hop | Temporal | Common | Avg |
| Full KV | – | 36.2 | 19.2 | 16.6 | 59.3 | 44.1 | 38.7 | 52.3 | 43.4 | 45.3 |
| SnapKV (Li et al., 2024) | 2K | 17.6 | 5.9 | 12.6 | 14.7 | 13.3 | 9.8 | 10.5 | 14.2 | 10.8 |
| | 4K | 23.8 | 8.0 | 13.1 | 24.2 | 20.1 | 15.8 | 16.0 | 24.8 | 17.2 |
| | 6K | 27.8 | 9.1 | 15.2 | 30.2 | 24.4 | 20.6 | 24.4 | 29.2 | 23.5 |
| | 8K | 30.8 | 13.2 | 14.9 | 39.4 | 30.8 | 24.4 | 25.1 | 33.5 | 26.1 |
| InfiniPot (Kim et al., 2024) | 2K | 15.1 | 14.2 | 12.0 | 12.4 | 13.2 | 10.4 | 15.7 | 15.2 | 13.4 |
| | 4K | 19.2 | 20.0 | 10.8 | 19.9 | 19.2 | 17.1 | 22.7 | 23.7 | 20.5 |
| | 6K | 23.4 | 23.7 | 15.0 | 26.0 | 24.3 | 21.6 | 30.0 | 25.5 | 25.8 |
| | 8K | 27.7 | 14.0 | 14.0 | 32.7 | 29.4 | 26.3 | 29.4 | 31.5 | 29.0 |
| KeyDiff (Park et al., 2025) | 2K | 12.8 | 16.7 | 12.4 | 13.6 | 14.0 | 9.8 | 15.0 | 18.4 | 13.3 |
| | 4K | 18.9 | 22.0 | 13.9 | 21.8 | 20.8 | 12.0 | 19.9 | 26.3 | 19.5 |
| | 6K | 26.3 | 25.2 | 14.3 | 29.8 | 27.2 | 15.7 | 26.1 | 21.1 | 21.1 |
| | 8K | 28.3 | 23.7 | 15.9 | 36.4 | 30.8 | 20.3 | 32.7 | 28.0 | 26.9 |
| KVzip (Kim et al., 2025) | 2K | 14.3 | 13.7 | 11.7 | 12.9 | 13.3 | 12.2 | 10.8 | 23.9 | 13.3 |
| | 4K | 19.4 | 16.2 | 13.9 | 20.6 | 19.0 | 17.7 | 16.0 | 29.8 | 18.8 |
| | 6K | 24.3 | 19.5 | 12.5 | 27.2 | 24.2 | 22.5 | 24.6 | 32.6 | 24.9 |
| | 8K | 25.6 | 23.8 | 13.0 | 34.8 | 29.5 | 26.8 | 28.7 | 37.7 | 29.3 |
| EPICACHE | 2K | 26.4 | 15.4 | 13.2 | 29.3 | 24.9 | 24.1 | 21.3 | 36.6 | 24.7 |
| | 4K | 29.0 | 20.9 | 14.1 | 38.5 | 31.6 | 31.1 | 29.3 | 37.5 | 31.3 |
| | 6K | 32.6 | 24.6 | 15.1 | 46.5 | 37.5 | 33.0 | 39.4 | 40.6 | 36.9 |
| | 8K | 32.6 | 28.1 | 15.3 | 52.7 | 41.6 | 33.8 | 46.9 | 43.6 | 41.0 |

Table A3: **LongConvQA (LoCoMo and Realtalk) Evaluation**: Comparison of different KV cache compression methods under block-prefill with Qwen series models.

| Method | $M$ | LLaMA3.2-3B | | | | | | | | LLaMA3.1-8B | | | | | | | |
|---|---|---|---|---|---|---|---|---|---|---|---|---|---|---|---|---|---|
| | | SH | TH | MS | TR-E | TR-I | KU | IP | Avg. | SH | TH | MS | TR-E | TR-I | KU | IP | Avg. |
| Full KV | 21K | 84.6 | 10.0 | 12.5 | 47.9 | 27.1 | 52.3 | 6.2 | 39.4 | 87.2 | 14.1 | 17.6 | 56.5 | 28.5 | 56.1 | 6.3 | 43.1 |
| SnapKV (Li et al., 2024) | 2K | 26.9 | 0.8 | 1.8 | 26.6 | 17.4 | 31.4 | 6.1 | 17.7 | 35.6 | 3.8 | 3.3 | 29.8 | 23.3 | 25.4 | 8.8 | 20.2 |
| | 4K | 40.0 | 5.3 | 2.4 | 40.2 | 20.4 | 48.5 | 6.5 | 26.1 | 63.3 | 9.7 | 3.9 | 45.8 | 24.4 | 45.8 | 10.8 | 32.4 |
| | 6K | 54.5 | 5.3 | 15.8 | 37.1 | 23.0 | 58.7 | 6.3 | 33.0 | 65.1 | 8.3 | 9.0 | 49.4 | 30.4 | 50.5 | 11.8 | 35.6 |
| | 8K | 67.1 | 7.9 | 12.4 | 37.1 | 27.1 | 56.6 | 7.1 | 35.6 | 74.3 | 13.1 | 13.6 | 53.8 | 25.9 | 52.5 | 10.9 | 38.7 |
| InfiniPot (Kim et al., 2024) | 2K | 46.2 | 0.8 | 10.8 | 40.5 | 19.1 | 40.7 | 8.9 | 26.3 | 39.8 | 1.8 | 3.4 | 29.1 | 31.7 | 35.8 | 7.6 | 24.1 |
| | 4K | 48.5 | 7.9 | 12.3 | 33.6 | 18.4 | 52.3 | 7.6 | 29.4 | 62.4 | 7.9 | 4.1 | 53.9 | 23.0 | 46.8 | 9.2 | 32.7 |
| | 6K | 60.0 | 2.6 | 12.4 | 33.6 | 25.9 | 51.7 | 6.7 | 31.9 | 81.3 | 11.0 | 13.3 | 54.9 | 28.1 | 54.9 | 5.6 | 40.4 |
| | 8K | 76.0 | 4.4 | 13.3 | 40.7 | 25.0 | 52.9 | 7.6 | 36.2 | 90.3 | 9.1 | 18.8 | 54.0 | 28.5 | 56.8 | 9.4 | 42.2 |
| KeyDiff (Park et al., 2025) | 2K | 35.3 | 0.5 | 4.1 | 34.3 | 17.7 | 5.7 | 2.6 | 15.1 | 26.0 | 6.8 | 9.5 | 28.8 | 12.3 | 22.8 | 5.5 | 17.1 |
| | 4K | 54.2 | 2.1 | 2.4 | 34.3 | 15.4 | 34.4 | 6.8 | 24.2 | 60.0 | 14.7 | 12.9 | 41.4 | 20.2 | 31.8 | 6.1 | 29.6 |
| | 6K | 55.8 | 6.5 | 7.4 | 54.3 | 11.4 | 32.2 | 9.0 | 26.9 | 69.2 | 11.2 | 13.6 | 48.4 | 26.1 | 52.1 | 8.5 | 36.7 |
| | 8K | 56.1 | 2.1 | 6.6 | 37.1 | 25.9 | 37.7 | 7.7 | 27.9 | 70.7 | 11.2 | 18.4 | 46.1 | 30.0 | 50.1 | 5.2 | 37.6 |
| KVzip (Kim et al., 2025) | 2K | 30.8 | 0.0 | 1.8 | 30.9 | 15.2 | 30.3 | 7.6 | 18.2 | 33.8 | 7.5 | 8.0 | 36.0 | 19.4 | 27.2 | 9.1 | 21.5 |
| | 4K | 44.7 | 2.6 | 6.5 | 37.1 | 15.0 | 37.8 | 7.4 | 24.0 | 56.6 | 9.7 | 6.7 | 35.3 | 24.1 | 42.7 | 12.4 | 29.8 |
| | 6K | 58.1 | 5.3 | 12.4 | 37.1 | 21.3 | 50.8 | 6.2 | 31.3 | 61.3 | 11.8 | 6.9 | 42.7 | 29.4 | 47.5 | 11.3 | 33.6 |
| | 8K | 73.5 | 7.9 | 12.5 | 40.7 | 26.2 | 59.1 | 6.9 | 37.5 | 73.6 | 14.4 | 7.7 | 48.6 | 26.9 | 51.4 | 6.2 | 37.1 |
| **EPICACHE** | 2K | 73.0 | 10.5 | 7.4 | 40.5 | 21.0 | 50.8 | 6.0 | 34.4 | 72.3 | 16.7 | 3.3 | 46.5 | 24.0 | 56.4 | 10.5 | 37.1 |
| | 4K | 79.9 | 12.6 | 16.6 | 41.4 | 27.0 | 53.9 | 9.1 | 39.3 | 87.2 | 14.1 | 17.4 | 56.5 | 28.5 | 54.6 | 3.9 | 42.6 |
| | 6K | 85.0 | 10.0 | 13.4 | 40.7 | 27.2 | 55.1 | 8.3 | 39.6 | 83.8 | 13.8 | 25.4 | 55.7 | 25.3 | 54.9 | 10.7 | 42.9 |
| | 8K | 85.0 | 10.0 | 12.5 | 40.7 | 26.6 | 56.6 | 6.2 | 39.5 | 88.2 | 13.5 | 17.5 | 56.5 | 28.5 | 55.2 | 6.4 | 43.0 |

| Method | $M$ | Qwen2.5-3B | | | | | | | | Qwen2.5-7B | | | | | | | |
|---|---|---|---|---|---|---|---|---|---|---|---|---|---|---|---|---|---|
| | | SH | TH | MS | TR-E | TR-I | KU | IP | Avg. | SH | TH | MS | TR-E | TR-I | KU | IP | Avg. |
| Full KV | 21K | 80.8 | 14.0 | 15.0 | 50.2 | 23.7 | 59.0 | 9.1 | 40.7 | 88.6 | 39.7 | 35.1 | 32.9 | 35.4 | 47.9 | 12.9 | 46.9 |
| SnapKV (Li et al., 2024) | 2K | 23.5 | 1.2 | 1.3 | 11.9 | 0.1 | 25.6 | 3.8 | 12.7 | 19.9 | 4.9 | 8.3 | 33.1 | 21.5 | 31.1 | 8.9 | 19.3 |
| | 4K | 48.8 | 6.5 | 0.8 | 11.9 | 1.1 | 27.9 | 4.6 | 18.7 | 42.9 | 10.9 | 22.0 | 33.6 | 28.8 | 38.4 | 10.7 | 29.3 |
| | 6K | 59.3 | 3.9 | 19.6 | 32.4 | 5.9 | 34.5 | 5.7 | 26.4 | 50.4 | 10.9 | 17.0 | 32.9 | 28.4 | 42.6 | 11.3 | 30.7 |
| | 8K | 56.3 | 11.4 | 20.4 | 29.8 | 7.9 | 41.7 | 8.1 | 28.6 | 67.4 | 17.1 | 24.4 | 32.9 | 31.6 | 42.9 | 14.1 | 36.7 |
| InfiniPot (Kim et al., 2024) | 2K | 31.0 | 1.2 | 11.9 | 22.6 | 5.4 | 17.5 | 4.9 | 14.8 | 29.8 | 1.6 | 2.0 | 33.3 | 23.1 | 32.2 | 11.8 | 20.5 |
| | 4K | 46.2 | 5.3 | 16.5 | 33.3 | 19.4 | 31.5 | 6.8 | 25.2 | 44.7 | 22.6 | 25.3 | 36.4 | 36.7 | 41.6 | 10.3 | 34.0 |
| | 6K | 55.9 | 4.8 | 10.6 | 32.4 | 25.7 | 40.1 | 6.1 | 28.7 | 56.4 | 26.7 | 32.9 | 36.4 | 40.5 | 48.1 | 11.7 | 41.7 |
| | 8K | 70.2 | 6.1 | 15.7 | 35.9 | 27.3 | 47.3 | 7.7 | 34.5 | 78.6 | 34.0 | 33.6 | 32.9 | 41.4 | 50.1 | 13.1 | 45.5 |
| KeyDiff (Park et al., 2025) | 2K | 12.6 | 14.0 | 6.3 | 44.1 | 11.1 | 27.6 | 3.3 | 17.3 | 18.1 | 8.2 | 9.4 | 31.1 | 14.3 | 31.5 | 11.7 | 18.4 |
| | 4K | 30.5 | 11.4 | 11.1 | 34.5 | 18.9 | 33.0 | 5.7 | 22.5 | 43.1 | 5.8 | 9.9 | 37.1 | 20.5 | 32.7 | 10.7 | 24.7 |
| | 6K | 51.6 | 15.3 | 21.1 | 43.7 | 20.5 | 35.4 | 9.6 | 30.4 | 57.5 | 13.0 | 13.0 | 34.2 | 26.8 | 38.0 | 9.9 | 30.8 |
| | 8K | 62.6 | 22.6 | 15.4 | 30.7 | 22.0 | 47.5 | 7.8 | 33.9 | 69.6 | 23.6 | 32.9 | 39.1 | 25.6 | 50.2 | 11.9 | 40.2 |
| KVzip (Kim et al., 2025) | 2K | 18.4 | 1.2 | 1.3 | 13.7 | 10.9 | 22.5 | 4.2 | 11.8 | 21.1 | 4.8 | 2.4 | 22.8 | 19.8 | 27.1 | 8.9 | 16.5 |
| | 4K | 39.4 | 6.5 | 5.0 | 23.8 | 13.8 | 25.3 | 6.4 | 19.2 | 45.4 | 14.1 | 22.0 | 31.7 | 23.3 | 29.9 | 11.5 | 27.5 |
| | 6K | 51.9 | 1.2 | 20.0 | 44.1 | 17.6 | 34.3 | 5.7 | 27.5 | 57.8 | 23.5 | 22.8 | 33.6 | 31.9 | 34.4 | 11.2 | 33.8 |
| | 8K | 68.4 | 8.8 | 15.0 | 39.5 | 20.7 | 40.9 | 10.0 | 32.5 | 69.6 | 23.5 | 23.4 | 32.9 | 35.0 | 40.3 | 10.6 | 37.7 |
| **EPICACHE** | 2K | 52.6 | 3.5 | 10.0 | 32.4 | 22.9 | 46.7 | 6.7 | 28.7 | 70.4 | 36.9 | 31.1 | 42.4 | 35.9 | 48.5 | 12.4 | 43.7 |
| | 4K | 74.4 | 14.0 | 11.0 | 46.7 | 17.6 | 55.9 | 9.1 | 37.0 | 83.6 | 38.3 | 29.8 | 37.6 | 49.6 | 41.3 | 12.8 | 46.6 |
| | 6K | 77.3 | 16.7 | 15.4 | 46.7 | 22.0 | 55.3 | 10.5 | 39.2 | 86.1 | 38.3 | 33.1 | 32.9 | 39.9 | 48.8 | 12.7 | 46.9 |
| | 8K | 77.3 | 16.7 | 15.0 | 46.7 | 22.8 | 55.9 | 10.1 | 39.4 | 86.0 | 38.3 | 33.1 | 37.6 | 40.5 | 47.3 | 12.9 | 47.2 |

Table A4: **LongConvQA (LongMemEval) Evaluation**: Evaluation results with Qwen and LLaMA series models under block prefill. SH = Single Hop, TH = Two Hop, MS = Multi-Session, TR-E = Temporal Reasoning (explicit), TR-I = Temporal Reasoning (implicit), KU = Knowledge Update. IP = Implicit Preference.

