# OpenReview forum: "EpiCache: Episodic KV Cache Management for Long Conversational Question Answering"
_ICLR.cc/2026/Conference — Submitted to ICLR 2026_

### Official Review · Reviewer_xjth · 2025-10-14

**Soundness:** 3
**Presentation:** 3
**Contribution:** 2
**Rating:** 4
**Confidence:** 5

**Summary:**

This paper introduces EPICACHE, a training-free framework for managing the Key-Value (KV) cache in large language models (LLMs) to enable long conversational question answering (LongConvQA) under strict, fixed memory budgets. The core problem it addresses is the linear growth of the KV cache with context length, which becomes a bottleneck in resource-constrained environments.

**Strengths:**

Significance of the Problem: The paper addresses a critical and timely problem in deploying LLMs: the memory and computational cost associated with very long contexts. The focus on LongConvQA under fixed memory budgets is highly relevant for practical applications like AI assistants on personal devices.



Novel Methodology: The core idea of combining block-wise prefill with episodic clustering is novel and well-motivated. Instead of treating the context as a monolithic block, partitioning it into semantically coherent topics is an intuitive and powerful approach for managing conversational memory. The framework is also training-free, which is a significant advantage for practical adoption.

**Weaknesses:**

Coarse Granularity and Untested Failure Modes: The method's core mechanism relies on retrieving a coarse-grained "episode-level" cache. This design may be insufficient for tasks requiring fine-grained information retrieval or cross-topic reasoning. For instance, it is unclear how EPICACHE would handle complex queries that need to synthesize details from multiple distinct episodes within the same conversation. Furthermore, its coarse granularity might cause it to fail on popular long-context benchmarks like Needle-in-a-Haystack (NIAH), which test precise fact retrieval, or coding tasks from benchmarks like LongBench, which require understanding fine-grained dependencies across a large codebase. The evaluation is limited to conversational QA and does not explore these likely failure points.



Lack of Batching Support and Deployment Inefficiency: The paper's efficiency analysis focuses exclusively on per-turn latency for a single user. The proposed method, which requires dynamically retrieving a different, query-specific cache for each request, is fundamentally incompatible with traditional batching techniques used in production environments. This lack of batching support means that while it may be efficient for a single user, it would be highly inefficient for actual deployment in a multi-user service, leading to low overall throughput.


Scalability and Hyperparameter Sensitivity: The episodic clustering approach relies on a fixed number of clusters, E (set to 4 in all experiments). It is unclear how this approach would scale to extremely long and diverse conversations that might contain dozens of distinct topics. The process would either lead to very broad, less useful clusters or require a much larger E, which would increase the offline storage cost and potentially the query-matching overhead. The paper lacks a sufficient analysis of the sensitivity to this crucial hyperparameter.

Insufficient Engagement with Concurrent Literature: The paper's literature review appears to be missing several highly relevant and concurrent works, making it difficult to fully situate its contribution. Specifically, the following works should be discussed and compared against:


* Cartridges [1], which proposes an alternative offline training approach (SELF-STUDY) to distill a large corpus into a small, parameterized KV cache.

* OracleKV [2] and ChunkKV [3], which explore semantic-aware compression techniques.

* FlowKV [4], which specifically addresses multi-turn conversational coherence, a goal very similar to EPICACHE's.

Reference:
> [1] Eyuboglu, Sabri, et al. "Cartridges: Lightweight and general-purpose long context representations via self-study." arXiv preprint arXiv:2506.06266 (2025).
> [2] Zhu, Yuanbing, et al. "OracleKV: Oracle Guidance for Question-Independent KV Cache Compression." ICML 2025 Workshop on Long-Context Foundation Models. 2025.
> [3] Liu, Xiang, et al. "Chunkkv: Semantic-preserving kv cache compression for efficient long-context llm inference." arXiv preprint arXiv:2502.00299 (2025).
> [4] Liu, Xiang, et al. "FlowKV: Enhancing Multi-Turn Conversational Coherence in LLMs via Isolated Key-Value Cache Management." arXiv preprint arXiv:2505.15347 (2025).

**Questions:**

See the weakness section.

---

> ### Author Response · Authors · 2025-11-27
>
> We sincerely appreciate it for the reviewer's constructive and insightful feedback.
>
> ---
>
> **[W1] Coarse Granularity and Untested Failure Modes**
>
> **W1-a. Contextualized Episodic KV Caches Enable Cross-Episode Reasoning**
>
> A key strength of episodic KV cache compression in EpiCache is that block-wise prefill encodes contextualized KV representations spanning the full dialogue, allowing fine-grained semantic interactions across episodes. This is validated by comparing against a naïve RAG-like episode-level retrieval method. In this RAG-like design, the conversation histroy is first clustered into episodes, and each cluster’s conversation raw text segments are directly fed into the model to build an episode-specific KV cache. A query then selects the most relevant episodic KV cache for answering.
>
> |    Method   | Multi-Hop | Temporal | Common |  Avg |
> | :---: | ---: | -----: | ----: | ---: |
> |   Full KV   |      49.2 |     55.9 |   48.4 | 52.0 |
> | *RAG-like-6K |      28.4 |     24.5 |   32.5 | 28.0 |
> |  KeyDiff-6K |      29.6 |     25.3 |   33.3 | 28.2 |
> | EpiCache-6K |      **42.3** |     **50.5** |   **35.2** | **46.5** |
>
> **Table R1.** RealTalk comparison between EpiCache and naïve episode-level retrieval / prior KV compression with LLaMA-3.1-8B. (From Appendix Fig.A1-a)
>
> RAG-like compression isolates the text within a single episode, its KV cache lacks cross-episode links, leading to severe degradation on multi-hop and temporal reasoning (–20 to –30 points vs. Full KV).
>
> In contrast, EpiCache applies block-wise prefill over the entire conversation while applying compression, preserving semantic relationships across distant topics. Thus, when episodic KV caches are later retrieved and used for generation, cross-episode reasoning remains intact and robust (e.g., 28.4 → 42.3 on Multihop; 24.5 → 50.5 on Temporal).
>
> Therefore, block-wise prefill with episodic KV cache compression yields memory-bounded KV caches that retain fine-grained episode-cross information, rather than restricting reasoning to a coarse-grained single episode.
>
> **W1-b. Generalization Beyond LongConvQA**
>
> While tasks requiring precise token-level retrieval (e.g., NIAH, repository-level coding) are outside the primary scope of LongConvQA, we demonstrate that our sensitivity-aware layer budgeting generalizes effectively to broader long-context tasks.
>
> We evaluated Qwen3-3B on LongBench (NarrativeQA, HotpotQA, Musique) under memory-constrained block-wise prefill. As shown in Table R2, applying our budgeting strategy on top of existing compression baselines (InfiniPot, KVzip) consistently improves performance across all memory budgets (e.g., +3.83 points with KVzip at 4K). This confirms that our sensitivity-based allocation is a robust, orthogonal module that enhances KV cache efficiency beyond conversational scenarios.
>
> | Qwen3-3B|2K|4K|6K|8K| Full |
> | ----- | :---------: | :-----: | :------: | :-----: | :--------: |
> | InfiniPot|       21.53       |       29.42       |       33.19       |       34.09       |    36.24   |
> | **InfiniPot + Ours** | 25.06 (+3.53) | 31.37 (+1.95) | 33.52 (+0.33) | 34.42 (+0.33) |    36.24   |
> | KVzip|       23.37       |       26.28       |       30.17       |       33.11       |    36.24   |
> | **KVzip + Ours**     | 24.49 (+1.12) | 30.11 (+3.83) | 32.53 (+2.36) | 34.02 (+0.91) |    36.24   |
>
> **Table R2.** LongBench accuracy under memory-constrained block-wise prefill. “+Ours” denotes applying our sensitivity-aware budget allocation on top of InfiniPot [R1] and KVzip [R2].
>
> ---
>
> **[W2] Lack of Batching Support and Deployment Inefficiency**
>
> As highlighted in the reviewer’s strength of our work, EpiCache is primarily designed for resource-constrained, on-device applications such as personal AI assistants. In these single-user scenarios, the critical objectives are minimizing peak memory usage to fit within limited hardware (e.g., ~10GB) and ensuring low latency for interactive responses, rather than maximizing multi-user throughput. Therefore, we focused our optimization on memory-bounded, single-batch inference to enable private, on-device execution of long-context models. However, we fully agree with the reviewer that adapting EpiCache for server-side batching is a crucial direction to enhance scalability, and we intend to address this in future work.

---

> ### Author Response · Authors · 2025-11-27
>
> **[W3] Scalability and Hyperparameter Sensitivity**
>
> | Method / Budget | 2K | 4K | 6K | 8K |
> | :--- | :---: | :---: | :---: | :---: |
> | KVzip (Baseline) | 21.2 | 31.4 | 39.1 | 44.8 |
> | Ours ($E=4$) | 33.2 | 41.2 | 45.8 | 47.4 |
> | Ours ($E=6$) | 35.9 | 43.1 | 46.3 | 48.1 |
> | Ours ($E=8$) | 37.4 | 43.7 | 46.6 | 48.7 |
>
> **Table R3.** LoCoMo score across number of episodes ($E$) with LLaMA3.1-8B (From Appendix Fig.A1-e)
>
> We clarify that our choice of a fixed cluster count ($E=4$) throughout the experiments is a design choice to ensure strictly bounded memory usage (including offline storage) for resource-constrained on-device environments. To validate this choice, we conducted a cluster number exploration on the LoCoMo benchmark, as detailed in Table R3 (data from Appendix Figure A.2)
>
> EpiCache ($E=4$) already significantly outperforms the baseline (e.g., 21.2 vs 33.2 at 2K budget). While increasing $E$ to 8 yields marginal gains, the performance tends to saturate as the KV budget increases.
>
>
> | Method / Length | 20K | 40K | 60K | 80K | 100K |
> | :--- | :---: | :---: | :---: | :---: | :---: |
> | Full KV | 39.4 | 35.3 | 31.7 | 31.4 | 31.6 |
> | InfiniPot (6K) | 31.9 | 28.7 | 19.4 | 13.3 | 14.3 |
> | KVzip (6K) | 31.3 | 20.5 | 14.3 | 11.8 | 10.5 |
> | EpiCache (6K) | 39.6 | 32.3 | 26.1 | 19.5 | 19.1 |
> | EpiCache (16K) | 39.7 | 35.5 | 30.1 | 29.0 | 29.3 |
> | EpiCache (24K) | 39.7 | 35.5 | 31.2 | 31.3 | 31.0 |
>
> **Table R4.** LongMemEval Accuracy across context lengths with LLaMA3.2-3B. (From Figure 6)
>
> To further demonstrate scalability to extremely long contexts, we evaluated LLaMA3.2-3B on LongMemEval across context lengths from 20K to 100K. As shown in Table R4, even with the fixed cluster count ($E=4$), EpiCache demonstrates superior robustness compared to baselines.
>
> At 100K context, baseline compression methods degrade severely to ~10-14% accuracy, EpiCache (16K) recovers the accuracy to 29.3%, nearly matching the Full KV performance (31.6%). This indicates that even with $E=4$, increasing the KV budget allows EpiCache to capture fine-grained details across the extensive conversation history, achieving near-lossless performance with a high compression ratio of $\sim$6$\times$ (16K/100K).
>
> Following the reviewer's suggestion, we will also include a detailed sensitivity analysis of episode counts ($E$) for 100K-scale scenarios in the revised manuscript.
>
> ---
>
> **[W4] Discussion with Concurrent Literature**
>
> We sincerely thank the reviewer for highlighting these relevant concurrent works. We will incorporate a detailed discussion and performance comparison in the revised manuscript. Below, we clarify the key distinctions and comparative advantages of EpiCache.
>
> - **Catridge**: While Cartridges effectively compresses context via "self-study," it is fundamentally a training-based approach requiring computation and data resource. In contrast, EpiCache is a training-free, plug-and-play framework, allowing immediate deployment on off-the-shelf LLMs without parameter updates.
> - **ChunkKV**: This method focuses on semantic chunking and layer-wise reuse. We consider this orthogonal to our approach; EpiCache manages episodic-level structures, while ChunkKV optimizes low-level chunk semantic selection. Integrating ChunkKV into our block-wise prefill pipeline is a promising direction for future work to further enhance answer quality in LongConvQA.
> - **FlowKV**: FlowKV manages short multi-turn coherence but has no evaluation on long conversation history. Its recent-turn eviction design cannot preserve fine-grained episode details scattered across long timelines, limiting applicability to LongConvQA under fixed memory.
> - **OracleKV**: OracleKV is the most relevant baseline as it also utilizes a patched-prompt mechanism for KV compression. However, it relies on a general prompt design derived from user prompt analysis. In contrast, EpiCache dynamically constructs episode-specific patched prompts, enabling far more precise context reconstruction for diverse conversational topics.
>
> To validate this, we compared EpiCache against OracleKV on RealTalk using Qwen3-4B-Instruct.
>
> | Method|2K |4K |6K |8K | Full |
> | :-- | --: | --: | --: | --: | --: |
> | KVzip|18.5 |23.9 |31.3 |35.9 | 56.9 |
> | SnapKV|16.7 |23.5 |28.5 |35.0 | 56.9 |
> | KeyDiff|7.0 |14.6 |24.7 |30.7 | 56.9 |
> | OracleKV|14.6 |25.4 |33.0 |38.7 | 56.9 |
> | EpiCache | **35.8** | **43.7** | **48.7** | **53.8** | 56.9 |
>
> **Table R6.** RealTalk Accuracy comparison including OracleKV (Qwen3-4B-Instruct)
>
> As shown in Table R6, while OracleKV outperforms previous baselines (e.g., 38.7 vs. 35.0 for SnapKV at 8K), it still falls significantly short of EpiCache (53.8). This substantial gap highlights the necessity of episode-aware management in LongConvQA, where a single general prompt cannot capture the distinct, multi-session conversational topics.
>
> ---
>
> We hope our responses have addressed your concerns. If anything remains unclear, we would be happy to provide further clarification.

---

### Official Review · Reviewer_tiys · 2025-10-25

**Soundness:** 2
**Presentation:** 2
**Contribution:** 2
**Rating:** 2
**Confidence:** 5

**Summary:**

This paper proposes EPICACHE, a training-free Key–Value (KV) cache management framework aimed at improving memory efficiency in long conversational question answering (LongConvQA). The key innovation lies in combining: Block-wise prefill for bounded memory growth, Episodic KV compression based on semantic clustering of conversation history, and Sensitivity-aware layer-wise budget allocation for better per-layer cache distribution. EPICACHE achieves up to 40% higher accuracy than existing methods (e.g., KVzip, InfiniPot) under fixed cache budgets and sustains near-full KV performance under 4–6× compression. Experiments span multiple LongConvQA benchmarks (Realtalk, LoCoMo, LongMemEval) and models (LLaMA, Qwen).

**Strengths:**

1. Evaluations across three benchmarks and four LLMs demonstrate robustness. EPICACHE consistently outperforms recent baselines.

2. Using unsupervised clustering to segment dialogue into semantically coherent “episodes” is intuitive for conversational QA.

**Weaknesses:**

1. Stage 1 of EPICACHE performs semantic segmentation and clustering of the full conversation history using a sentence encoder. While the paper describes this as “offline” and excludes it from latency measurements (Figure 7a), this is not realistic for many LongConvQA use cases, where conversations evolve dynamically and the clustering may need to be updated online.
- Cost not reflected in latency (TTFT): The encoder and K-Means clustering operations are non-trivial, especially when histories exceed tens of thousands of utterances. The cost should contribute to time-to-first-token (TTFT) or system-level latency, since users cannot interact until episodic caches are prepared.

2. Stage 2 performs episodic, query-independent KV eviction, meaning each episode cache is built once using the medoid segment as a static patched prompt. This design ensures bounded memory but raises two serious concerns:
- Because eviction decisions are made offline and not conditioned on actual upcoming queries, the cache may not capture relevant tokens for unseen topics or cross-episode reasoning. The approach could fail when conversations shift topics abruptly or when queries require combining information across episodes.
- In natural multi-turn dialogues, new utterances and model-generated responses continuously extend the context. The paper does not clarify: 1. Whether these new turns are inserted into existing episode clusters or trigger re-clustering. 2. Whether episodic caches are periodically rebuilt or incrementally updated. If the clustering remains static, cache quality will degrade over time as the conversation drifts semantically from initial clusters. If frequent re-clustering is required, the overhead could become prohibitive.

3. The proposed sensitivity-aware layer budgeting (Eq. 10) is empirically effective but seems somewhat ad hoc; an ablation isolating this contribution is only briefly mentioned.

**Questions:**

1. How expensive is Stage 1 in realistic settings (e.g., a 20 K–50 K token conversation)? Does this cost contribute to TTFT or total inference latency?

2. Can the clustering be performed incrementally or asynchronously to amortize cost?

3. How does EPICACHE handle newly generated tokens in an ongoing conversation? Are they appended to existing clusters? Does the system periodically rebuild episodic caches?

---

> ### Author Response · Authors · 2025-11-27
>
> We sincerely appreciate it for the reviewer's constructive and insightful feedback.
>
> **[W1, Q1,2] Latency Cost of KV Cache Build Stage in EpiCache**
> > W1) Stage 1 of EPICACHE is not realistic for many LongConvQA use cases, where conversations evolve dynamically and the clustering may need to be updated online.
> Q1, 2) How expensive is Stage 1 in realistic settings? Does this cost contribute to TTFT or total inference latency? Can the clustering be performed incrementally or asynchronously to amortize cost?
>
> EpiCache addresses multi-day conversational histories where topics change frequently across sessions. To preserve topic-relevant information under a fixed memory budget, we segment the history into semantically coherent episodes and build episodic KV caches as illustrated in Fig. 3. To construct these episodic KV cache sets, we perform the following three lightweight steps:
>
> - 1) sentence embedding of the conversation history, (Embed in Table R1)
> - 2) semantic clustering (Cluster in Table R1)
> - 3) block-wise prefill for KV cache construction (Prefill in Table R1)
>    - One prefill per episode (e.g., |E| = 4, 4 block-wise prefill required)
>
> After this one-time preparation (Prep. Total) for constructing episodic KV cache, the model reuses the these to answer subsequent queries (Decoding) without any additional prefill stage.
>
> We measure the latency of this preparation stage and report how much it contributes to total inference latency in realistic LongConvQA scenarios. Because a single conversational session contains hundreds of dependent user queries (~200 QAs in RealTalk), the preparation cost should be amortized over the entire interaction (Amorized Prep.). Below, we present the effective per-query latency by dividing the one-time cost by the number of QA turns.
>
> | Method                      | Embed | Cluster |    Prefill   | Prep. Total | Amortized Prep. | Decoding | Amortized Prep.+Dec. | Peak GPU Mem | Realtalk Score |
> | --------------------------- | :-------: | :---------: | :--------------: | :-------------: | :--------------------: | :-------------: | :---------------: | :---------------: | :--------------: |
> | Full KV                     |     0     |      0      |       8.6       |       8.6      |          0.04          |       0.7      |        0.7       |        27.4       |        40.3        |
> | Block-wise KV Cache Manage. |     0     |      0      |       8.2       |       8.2      |          0.04          |       0.3      |        0.3       |        8.9       |        25.0        |
> | EpiCache (E=4)              |    8.2   |     0.3    | 33.6 |      42.2      |          0.2          |       0.3      |        0.5       |        8.9  |        38.3        |
>
> **Table R1**. Latency (sec), GPU Peak Memory (GB), Realtalk Accuracy in LongConvQA (200 QAs per conversation) with LLaMA3.2-3B (A6000 single GPU used for measurement)
>
> As shown in Table R1, EpiCache introduces a modest *one-time* preparation cost of 42.2 sec on RealTalk, which amortizes to only 0.2 sec per query.
> The resulting effective latency per QA turn (0.51 sec) is lower than Full KV decoding (0.74 sec), while achieving +13.3 points higher accuracy compared to existing block-prefill KV cache management methods.

---

> ### Author Response · Authors · 2025-11-27
>
> **[W2-1] Unseen Topics, Cross-Episode Reasoning**
> > Because eviction decisions are made offline and not conditioned on actual upcoming queries, the cache may not capture relevant tokens for unseen topics or cross-episode reasoning. The approach could fail when conversations shift topics abruptly or when queries require combining information across episodes.
>
> - **Unseen Topics**: In the LongConvQA formulation (Eq. 1), a query is always expected to refer to information previously grounded in the shared dialogue context. When a new topic that has never appeared is introduced, it can simply respond based on the newly provided user input. Thus, unseen-topic queries do not reflect long-term conversational memory retention under bounded memory, which is the focus of our setting.
> - **Cross-Episode Reasoning**: This is a great point. RealTalk includes multi-hop subtasks that explicitly evaluate cross-episode reasoning, where answering requires integrating information across distinct topics and timestamps. To assess this capability, we compare against a naive episode-level retrieval approach (RAG-like) and KV cache compression baseline method - KeyDiff [R1] under the same 6K KV budget:
>
> |    Method   | Multi-Hop | Temporal | Common |  Avg |
> | :---------: | --------: | -------: | -----: | ---: |
> |   Full KV   |      49.2 |     55.9 |   48.4 | 52.0 |
> | *RAG-like-6K |      28.4 |     24.5 |   32.5 | 28.0 |
> |  KeyDiff-6K |      29.6 |     25.3 |   33.3 | 28.2 |
> | EpiCache-6K |      **42.3** |     **50.5** |   **35.2** | **46.5** |
>
> **Table R2**. Realtalk comparison between EpiCache and RAG-like and KeyDiff with LLaMA-3.1-8B used. (From Appendix Fig.A1-a)
>
> (*RAG-like baseline: text-based clustered conversation segments are retrieved per query and directly consumed as input, rather than block-wise KV cache management.)
>
> EpiCache achieves 42.3 multi-hop accuracy under the same 6K KV budget, compared to 28.4 for RAG-like and 29.6 for KeyDiff, yielding a margin of roughly +14–15 points on the cross-episode reasoning subtask. This result indicates that although episodic caches are built per cluster, block-wise prefill operates over the full dialogue context when forming each episodic KV cache, enabling contextualized representations that support reasoning across episodes. In contrast, RAG-like approaches isolate only the text segments relevant to the current query, resulting in limited cross-episode reasoning capability.
>
> In other words, even when the answer generation conditions on a KV cache set specialized single episode, the Key-Value representation encoded during block-prefill stage preserves the semantic interactions, leading to accuracy close to Full KV and far above query-conditioned retrieval baselines.
>
> ---
>
> **[W2-2, Q-2,3] KV Cache Management for On-Going Conversation**
> >  W2-2. Whether these new turns are inserted into existing episode clusters or trigger re-clustering. Whether episodic caches are periodically rebuilt or incrementally updated.
> Q2, 3. Can the clustering be performed incrementally or asynchronously to amortize cost? How does EPICACHE handle newly generated tokens in an ongoing conversation? Are they appended to existing clusters? Does the system periodically rebuild episodic caches?
>
> In an on-going conversation, if a new turn is semantically aligned with an existing episode, its KV entries are appended to that episode’s cache; if it represents a new topic, a new episodic KV set is initialized and built up. In both cases, re-clustering/compression is triggered only when the allocated memory budget for the corresponding episode is exceeded. The systematic details are as follows.
>
> 1. High similarity (≥ τ): continuing an existing episode.
>    The system retrieves the episodic compressed KV cache of the best-matching episode, answers the query conditioned on that cache, and appends the new KV entries to the same episode. When accumulated KV exceeds the memory budget ($M + M_{block}$), we re-run block-wise prefill compression within that episode, updating the medoid for patched-prompt accordingly.
>
> 2. Low similarity (< τ): starting a new episode.
>    If no centroid is sufficiently similar, the turn indicates a new topic. We create a new KV cache set without retrieving previous caches. Once its KV size exceeds ($M + M_{block}$), this new set is compressed into a new episodic KV cache.
>
> Re-clustering is triggered based on the predefined memory budget. In both cases—whether a turn extends an existing episode or starts a new one—we incrementally perform clustering and compression once the KV cache size of that episode reaches the threshold M + M_block. Under the RealTalk setting with M=6K, M_block=2K, this results in additional clustering and compression being invoked approximately every 800 turns on average.

---

> ### Author Response · Authors · 2025-11-27
>
> **[W3] Sensitivity-aware Layer Budgeting Ablation Study**
> > w3. The proposed sensitivity-aware layer budgeting (Eq. 10) is empirically effective but seems somewhat ad hoc; an ablation isolating this contribution is only briefly mentioned.
>
> We thank the reviewer for highlighting the importance of clearly isolating the contribution of the proposed layer-wise budgeting. To directly evaluate its effectiveness, we conduct an ablation study that compares (i) clustering only and (ii) clustering combined with sensitivity-aware layer budgeting, alongside recent KV cache compression baselines on RealTalk using a new model, Qwen3-4B-Instruct-2507, across four memory budgets (M = 2K–8K).
>
> | Method                        |        2K |        4K |       6K |        8K | Full |
> | :---------------------------- | --------: | --------: | -------: | --------: | ---: |
> | KVzip                         |      18.5 |      23.9 |     31.3 |     35.9 | 56.9 |
> | SnapKV                        |      16.7 |      23.5 |     28.5 |      35.0 | 56.9 |
> | KeyDiff                       |      7.0 |     14.6 |    24.7 |     30.7 | 56.9 |
> | Ours – Clustering only        |     34.6 |     40.1 |     45.3 |     49.7 | 56.9 |
> | Ours – Clustering + Budgeting | **35.8** | **43.7** | **48.7** | **53.8** | 56.9 |
>
> **Table R3.** Ablation of sensitivity-aware layer budgeting on RealTalk (Qwen3-4B-Instruct-2507).
>
> - Clustering alone improves baseline methods by +10 to +20 points
> - Layer budgeting yields additional +3 to +4.5 points consistently across budgets
>
> The proposed budgeting consistently shows effectiveness that brings compressed KV performance closer to Full KV while maintaining the same memory bound. We will add these ablation results and expanded explanation to the revised version.
>
> ---
>
> We hope our responses have addressed your concerns. If anything remains unclear, we would be happy to provide further clarification.
>
> [R1] Park et al, "KeyDiff: Key Similarity-Based KV Cache Eviction for Long-Context LLM Inference in Resource-Constrained Environments" NeurIPS 25

---

### Official Review · Reviewer_Wyhg · 2025-11-02

**Soundness:** 3
**Presentation:** 3
**Contribution:** 2
**Rating:** 4
**Confidence:** 4

**Summary:**

This paper presents EPICACHE, a training-free framework for Key-Value (KV) cache management in large language models (LLMs) designed for Long Conversational Question Answering (LongConvQA) under fixed memory constraints.EPICACHE introduces (a) episodic KV compression, that clusters conversation histories into semantically coherent episodes for topic-aware caching, and (b) sensitivity-aware layer-wise budget allocation, which allocates cache budgets based on each layer's sensitivity to eviction. Experiments across three benchmarks (Realtalk, LoCoMo, and LongMemEval) and four LLMs (LLaMA-3 and Qwen2.5 variants) demonstrate up to 40% accuracy improvement and 3.5× memory reduction over baselines.

**Strengths:**

S1. This paper tackles an important problem of long conversation compression.

S2. The paper proposed a well structured way to find the importance of different layers, and allocate different budget to them.

S3. The paper is well written and structured.

**Weaknesses:**

W1. EpiCache is limited to multi-round conversation-based long context. It can not be directly applied to long context and long generation tasks.

W2. I am not fully convinced by the effectiveness of the embedding model that can group the semantically similar episodes together. My intuition is that the episodes in a same conversation is very similar or maybe even all under the same topics. So it is doubtable whether different episodes can be effectively grouped.

W3. The layer-wise sharpness hyperparameter alpha in equation 10 should also be studied in the sensitivity experiments.

W4. It would be more convincing if the authors can include the performance on a medium-size models, like 32B models. In addition, Qwen2.5 should be replace by the new Qwen3 models.

**Questions:**

Q1. If the conversation continues to group, how would you change the number of the clusters? Would you and how often would you re-cluster the episodes?

Q2. Can you play with different embedding models?

Q3. Can you also compare your approach with more recent sparse KV cache baselines, like IceCache, MagicPig, ArkVale, InfiniGen, etc? I understand some of them are not optimized for long prefilling, but most of them can run on single model of 8B models for up to ~100k context.

---

> ### Author Response · Authors · 2025-11-27
>
> We sincerely appreciate it for the reviewer's constructive and insightful feedback.
>
> **[W1] Applicability of EpiCache to General Long Context Tasks**
> > W1. EpiCache can not be directly applied to long context and long generation tasks.
>
> Our work is designed to address long conversation history based multi-turn scenarios, where maintaining topic-relevant context across multi-session dialogues is the core challenge. We agree that applying episodic KV management to long-document or long-generation tasks is an interesting direction for future work.
>
> Our sensitivity-aware layer-wise budget allocation applies orthogonally to general long-context tasks under the memory-constrained block-wise prefill. To demonstrate this, we additionally evaluated on Qwen3-3B-Instruct using three long-document QA tasks from LongBench (NarrativeQA, HotpotQA, Musique) as shown in Table R1. Under block-wise prefill, combining our allocation with existing compression methods improves performance across all KV budgets, confirming that our budgeting strategy generalizes beyond conversations to broader long-context tasks.
>
> | Qwen3-3B|2K|4K|6K|8K| Full |
> | ----- | :---------: | :-----: | :------: | :-----: | :--------: |
> | InfiniPot|       21.53       |       29.42       |       33.19       |       34.09       |    36.24   |
> | **InfiniPot + Ours** | 25.06 (+3.53) | 31.37 (+1.95) | 33.52 (+0.33) | 34.42 (+0.33) |    36.24   |
> | KVzip|       23.37       |       26.28       |       30.17       |       33.11       |    36.24   |
> | **KVzip + Ours**     | 24.49 (+1.12) | 30.11 (+3.83) | 32.53 (+2.36) | 34.02 (+0.91) |    36.24   |
>
> **Table R1.** LongBench accuracy under memory-constrained block-wise prefill. “+Ours” denotes applying our sensitivity-aware budget allocation on top of InfiniPot [R1] and KVzip [R2].
>
> ---
>
> **[W2] Validity of Episodic Clustering in Long Multi-Session Dialogues**
> > W2. I am not fully convinced by the effectiveness of the embedding model that can group the semantically similar episodes together.
>
> We appreciate the reviewer’s points. We clarify that a “single conversation” in our setting spans multi-day human–LLM interaction histories (16–35 days; 20K–100K tokens) in RealTalk / LoCoMo / LongMemEval. Such long-term dialogues naturally exhibit frequent topic shifts across days and sessions, and thus the intuition that all turns belong to a single topic does not hold in these realistic LongConvQA scenarios.
>
> Capturing these episodic transitions is the central challenge of bounded-memory conversational QA. We find that grouping the history into semantically coherent episodes and performing episode-specific KV cache compression is essential for preserving such topic-relevant information under strict memory budgets, and this design leads to substantial accuracy gains over non-episodic compression strategies.
>
> ---
>
> **[W3] Sharpness Hyperparameter (α) Exploration**
> > W3. The layer-wise sharpness hyperparameter alpha in equation 10 should also be studied in the sensitivity experiments.
>
> We thank the reviewer for highlighting this important hyperparameter. We conducted a sweep on the sharpness parameter ( $\alpha$ ) in Eq. (10) under the block-wise prefill setting (Qwen3-3B-Instruct, M=6K, RealTalk). As shown in Table R2, sensitivity-aware budgeting improves accuracy by ~5 points over the Uniform allocation baseline on both models, and values around ($\alpha \approx 1$) consistently yield the best accuracy:
>
> | Model | Uniform | 0.25 |  0.5 |  **1.0** |  2.0 |  4.0 |  6.0 |
> | ---------: | :-----: | :--: | :--: | :------: | :--: | :--: | :--: |
> |     Qwen3-3B |   45.3  | 47.1 | 48.0 | **50.6** | 50.3 | 49.5 | 49.1 |
> |     Qwen2.5-7B |   32.6  | 33.4 | 34.2 | **37.2** | 37.1 | 32.4 | 29.9 |
>
> **Table R2.** Ablation on the sharpness parameter used for sensitivity-aware layer budgeting.
>
> - Low sharpness ($\alpha$ $\le$ 0.5) : only mild differentiation across layers
> - Moderate sharpness ($\alpha$ $\approx$ $1{-}2$) : effectively promotes KV cache budget sensitivity-critical layers while maintaining sufficient budget for others
> - High sharpness ($\alpha$ $\ge$ 4) : the budget becomes overly concentrated on a few layers
>
> These results indicate that moderate sharpness effectively emphasizes layer sensitivity without overly concentrating the budget on a few layers. We will add these findings to the revised manuscript.

---

> ### Author Response · Authors · 2025-11-27
>
> **[W4] Model Expansion (Qwen3)**
> > Performance on a medium-size models, like 32B models. In addition, Qwen2.5 should be replace by the Qwen3 models.
>
> We appreciate the reviewer's suggestion. As shown in Table R3, we evaluate EpiCache to the latest Qwen3-3B-Instruct-2507 on RealTalk, and our method consistently outperforms recent KV compression baselines under all memory budgets. Larger (30B+) models are beyond our current deployment focus but are an exciting direction we plan to explore in future work.
>
> | Method|2K |4K |6K |8K | Full |
> | :-- | -: | -: | -: | -: | ---: |
> | KeyDiff|7.0 |14.6 |24.7 |30.7 | 56.9 |
> | KVzip|18.5 |23.9 |31.3 |35.9 | 56.9 |
> | SnapKV|16.7 |23.5 |28.5 |35.0 | 56.9 |
> | EpiCache | **35.8** | **43.7** | **48.7** | **53.8** | 56.9 |
>
> **Table R3.** Block-wise Prefill comparison with Qwen3-3B-Instruct under Realtalk benchmark
>
> ---
>
> **[Q1] Cluster Growth and Re-clustering Frequency**
> > Q1. If the conversation continues to group, how would you change the number of the clusters? Would you and how often would you re-cluster the episodes?
>
> In an on-going conversation, if a new turn is semantically aligned with an existing episode, its KV entries are appended to that episode’s cache; if it represents a new topic, a new episodic KV set is initialized and built up. In both cases, re-clustering/compression is triggered only when the allocated memory budget for the corresponding episode is exceeded. The systematic details are as follows.
>
> 1. High similarity (≥ τ): continuing an existing episode.
>    The system retrieves the episodic compressed KV cache of the best-matching episode, answers the query conditioned on that cache, and appends the new KV entries to the same episode. When accumulated KV exceeds the memory budget ($M + M_{block}$), we re-run block-wise prefill compression within that episode, updating the medoid for patched-prompt accordingly.
>
> 2. Low similarity (< τ): starting a new episode.
>    If no centroid is sufficiently similar, the turn indicates a new topic. We create a new KV cache set without retrieving previous caches. Once its KV size exceeds ($M + M_{block}$), we perform clustering to obtain a new medoids and compress the KV set into the memory budget $M$ using patched-prompt block-prefill.
>
> Re-Clustering is triggered based on the predefined memory budget. In both cases—whether a turn extends an existing episode or starts a new one—we incrementally perform clustering and compression once the KV cache size of that episode reaches the threshold M + M_block. Under the RealTalk setting with M=6K, M_block=2K, this results in additional clustering and compression being invoked approximately every 800 turns on average.
>
>
> ---
>
> **[Q2] Embedding Model Exploration**
> > Can you play with different embedding models?
>
> Yes. As reported in Fig. A1 (Alternative Design Exploration) and Table R4, we explored multiple embedding models for episodic clustering. Intuitively, dedicated sentence encoders (e.g., MiniLM, Qwen3-Embedding) outperform using LLM’s internal embedding layer, and larger encoders generally perform better. Notably, even a 60M model (MiniLM) already brings large gains over the baseline, showing the robustness of our design.
>
> | Method / KV Budget          |   2K  |   4K  |   6K  |   8K  |
> | -- | :---: | :---: | :---: | :---: |
> | InfiniPot (baseline)| 16.69 | 25.77 | 34.48 | 38.84 |
> | *LLM Embed| 25.65 | 36.48 | 42.52 | 46.06 |
> | EpiCache-MiniLM-L6-v2 | 32.10 | 40.20 | 44.90 | 47.90 |
> | EpiCache-Qwen3-0.6B   | 33.20 | 41.20 | 45.80 | 47.40 |
> | EpiCache-Qwen3-4B     | 33.20 | 40.70 | 45.80 | 48.10 |
>
> **Table R4.** LoCoMo F1 Score under block-wise prefill (Qwen2.5-7B), varying embedding models. (From Appendix Fig.A1-d)
>
> (*LLM Embed: text-embedding layer of the target LLM)
>
> ---
>
> **[Q3] Comparison with Recent Sparse KV Cache Baselines**
> > Can you also compare your approach with more recent sparse KV cache baselines, like IceCache, MagicPig, ArkVale, InfiniGen, etc?
>
> We thank the reviewer for highlighting these recent sparse-KV methods. Their primary focus is reducing the KV cache used during decoding, but they still require building and storing the *full* KV cache during the prefill phase. Thus, their memory consumption continues to scale with the conversation length.
>
> In contrast, EpiCache is designed for strictly bounded memory scenarios, where the KV cache is never allowed to grow beyond a fixed budget. Through block-wise prefill combined with episodic compression, memory remains constant regardless of context length (as shown in Figure 1(c)), which is the core problem setting of this work.
>
> While the scope and system assumptions differ, we agree that these techniques are relevant, and we will include comparative discussion—and evaluation where applicable—in the revised version.
>
> [R1] Kim et al, InfiniPot: Infinite Context Processing on Memory-Constrained LLMs, EMNLP 24
> [R2] Kim et al, kvzip: query-agnostic kv cache compression with context reconstruction, NeurIPS 25

---

### Meta-Review · Area_Chair_HcC2 · 2026-01-06

**Summary:**

The paper proposes EpiCache, a training-free KV-cache management framework for long conversational QA under fixed memory budgets, combining episodic clustering with layer-wise budget allocation. Reviewers agree the problem is important and the approach is reasonably well-motivated, but the main blockers are (i) unclear robustness/generalization beyond the specific LongConvQA setting, (ii) practicality concerns for deployment (e.g., batching incompatibility; system-level efficiency), and (iii) insufficient clarity/analysis around episodic clustering dynamics (scaling to long, evolving conversations; sensitivity to key hyperparameters).

**Reviewer Concerns:**

Addressed (partially) by rebuttal:

+Layer-wise budgeting contribution: authors provide a clearer ablation separating clustering vs. clustering+budgeting and show consistent gains.

+Stage-1 preparation cost: authors quantify embedding+clustering+prefill overhead and argue it amortizes over many QA turns in their target setting.

+Coarse “episode-level” concern / cross-episode reasoning: authors add an empirical comparison to a naïve episode-retrieval design and claim EpiCache preserves cross-episode interactions better

Still outstanding:

-Deployment realism & batching: a key critique is that query-specific cache retrieval is fundamentally at odds with standard batching/throughput in production; the rebuttal does not resolve this beyond per-session amortization arguments.

-Dynamic, evolving conversations: concerns remain about how clustering/caches are updated as new turns arrive (re-clustering frequency, incremental updates, degradation over time); the rebuttal discusses “one-time preparation” but does not fully settle the operational story across realistic evolving sessions.

-Scaling/sensitivity & missing comparisons/positioning: reviewers request stronger sensitivity analysis (e.g., cluster number, other embedding models, and other key hyperparameters) and better engagement with concurrent cache-compression literature.

**Reviewer Scores:**

Given the current rebuttal, I would not expect significant review score changes.

---

### Decision · Program_Chairs · 2026-01-26

Reject